# FedRTS: Federated Robust Pruning via Combinatorial Thompson Sampling

**Hong Huang, Jinhai Yang, Yuan Chen, Jiaxun Ye, Dapeng Wu**
Department of Computer Science
City University of Hong Kong
Hong Kong SAR, China
{hohuang-c, jinhayang7-c, ychen2752-c, jiaxunye-c}@my.cityu.edu.hk
dpwu@ieee.org

## Abstract

Federated Learning (FL) enables collaborative model training across distributed clients without data sharing, but its high computational and communication demands strain resource-constrained devices. While existing methods use dynamic pruning to improve efficiency by periodically adjusting sparse model topologies while maintaining sparsity, these approaches suffer from issues such as **greedy adjustments**, **unstable topologies**, and **communication inefficiency**, resulting in less robust models and suboptimal performance under data heterogeneity and partial client availability. To address these challenges, we propose **Fed**erated **R**obust pruning via combinatorial **T**hompson **S**ampling (FedRTS), a novel framework designed to develop robust sparse models. FedRTS enhances robustness and performance through its Thompson Sampling-based Adjustment (TSAdj) mechanism, which uses probabilistic decisions informed by stable and farsighted information, instead of deterministic decisions reliant on unstable and myopic information in previous methods. Extensive experiments demonstrate that FedRTS achieves state-of-the-art performance in computer vision and natural language processing tasks while reducing communication costs, particularly excelling in scenarios with heterogeneous data distributions and partial client participation. Our codes are available at: https://github.com/Little0o0/FedRTS.

## 1 Introduction

Federated learning (FL) [34, 30, 25, 64, 65] enables edge devices with private local data to collaboratively train a model without sharing data. However, substantial computational and communication demands for training deep learning models often exceed the resource limitations of edge devices in cross-device FL scenarios. Although neural network pruning [12, 40, 33] offers a promising solution by removing redundant parameters, traditional pruning methods depend on resource-intensive dense model training, rendering them impractical for privacy-preserved and resource-constrained FL environments.

To address these challenges, recent federated pruning frameworks [1, 47, 57, 24, 19, 18, 42] have adopted dynamic sparse training techniques [11, 48, 22] within FL. These frameworks employ a two-loop training process: in the inner loop, model weights are updated through standard FL rounds with fixed model topology; in the outer loop, the server adjusts the model topology by pruning and reactivating parameters [11], as illustrated in Fig. 1 (left). This iterative process generates a specialized sparse model without dense model training, significantly reducing computational costs.

Despite these advancements, existing frameworks [1, 47, 57, 24, 19, 18, 42] suffer from three critical challenges in model topology adjustment, as illustrated in Fig. 1 (left). **(1) Greedy adjustment:**

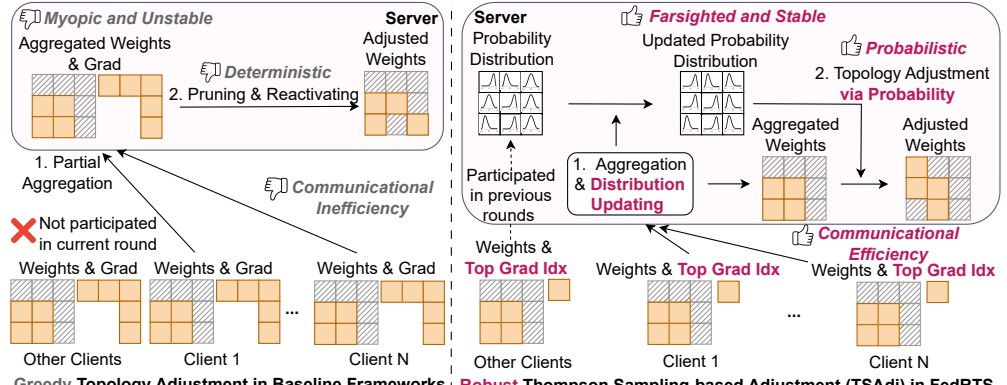

Figure 1: Illustration of topology adjustment in existing baselines [1] and FedRTS. *Left:* Existing baselines adjust model topology based on myopic and unstable aggregated weights and gradients via deterministic magnitude pruning and reactivating, resulting in greedy adjustment and high communication overhead. *Right:* FedRTS introduces Thompson Sampling-based Adjustment (TSAdj) to adjust the topology based on farsighted and stable probability distributions, achieving a robust topology and low communication overhead.

Current methods rely on myopic, aggregated information from a small subset of participating clients, ignoring data from the majority of unseen clients and prior knowledge. This leads to greedy adjustments and reduced robustness [25]. **(2) Unstable topology:** Deterministic adjustments based solely on aggregated information are prone to instability due to heterogeneous data distributions, resulting in unstable model topologies. **(3) Communication inefficiency:** Transmitting extensive auxiliary data (*e.g.,* full-size gradients) for topology updates imposes high communication costs. These limitations hinder the ability of current methods to handle client availability, data heterogeneity, and communication costs effectively, ultimately leading to suboptimal performance and inefficient resource utilization.

To address these challenges, *we reframe federated pruning as a combinatorial multi-armed bandit (CMAB) [5] problem and identify the above issues located in myopic observation and deterministic decision.* Building on this insight, we propose **Fed**erated **R**obust pruning via combinatorial **T**hompson **S**ampling (FedRTS), a novel framework designed to derive robust sparse models for cross-device FL, as illustrated in Fig. 2. FedRTS introduces the Thompson Sampling-based Adjustment (TSAdj) mechanism, depicted in Fig. 1 (right), to address the shortcomings of existing methods. First, TSAdj leverages farsighted probability distributions (including prior information from unseen clients) to mitigate the impact of partial client participation. Second, it uses probabilistic decisions based on stable, comprehensive information for topology adjustment. Third, FedRTS reduces communication overhead by requiring clients to upload only top-gradient indices instead of dense gradients.

We evaluate FedRTS on computer vision and natural language processing datasets, demonstrating its effectiveness in handling client availability and data heterogeneity. FedRTS achieves higher accuracy, better generalization, and lower communication costs compared to state-of-the-art (SOTA) methods. For instance, on the CIFAR-10 dataset with the ResNet18 model, FedRTS achieves either a 5.1% accuracy improvement or a 33.3% reduction in communication costs compared to SOTA frameworks. *To our knowledge, this paper is the first work to analyze federated pruning through a CMAB lens and apply combinatorial Thompson sampling to stabilize topology adjustments in FL.*

## 2 Preliminary and Challenges

### 2.1 Federated Dynamic Pruning

In federated pruning, $\mathcal{N}$ resource-constrained clients collaboratively train a sparse model using their local datasets $D_n, n \in \{1, 2, ..., \mathcal{N}\}$. Given the target density $d'$, the objective is to solve the

---

[1]Baseline's workflow: `https://github.com/Little0o0/FedRTS/tree/main/image/workflow.mp4`

following optimization problem:

$$\min_{W,m} \sum_{n=1}^{\mathcal{N}} p_n f_n(W, m, D_n), \quad \text{s.t. } d \le d', \tag{1}$$

where $W$ represents the model weights, $m \in \{0,1\}^{\langle W \rangle}$ denotes the sparse model topology (also called mask), $f_n(\cdot)$ is the objective function depending on the task of the $n$-th for client $n$ (*e.g.,* cross-entropy loss for the image classification task), $d = \frac{\text{sum}(m)}{\langle m \rangle}$ is the density of the topology $m$, and $p_n$ is set as the proportion of the data size of client $n$, typically set as $p_n = \langle D_n \rangle / \sum_{i=1}^{\mathcal{N}} \langle D_i \rangle$. [2]

To solve the problem in Eq. 1 under the target density constraint $d'$, existing federated pruning frameworks [1, 47, 57, 24, 19, 18, 42] apply dynamic sparse training techniques [11, 48, 22]. These frameworks iteratively adjust sparse on-device models during training while maintaining the density level $d \le d'$. The process begins with a sparse model initialized via random pruning and follows a two-loop training procedure: In the inner loop, the model topology $m$ remains fixed, and the model weights $W$ are updated through traditional FL rounds. After $\Delta T$ inner loops, as illustrated in Fig. 1 (left), the framework enters the outer loop, where clients upload additional data (specifically, the gradients of the pruned weights) to the server. The server then adjusts the model topology by pruning and reactivating [11] based on the aggregated weights and gradients.

## 2.2 CMAB-Based Problem Formulation

The Combinatorial Multi-Armed Bandit (CMAB) [5, 55, 26] is a sequential decision-making framework where an agent selects combinations of arms (called super arms) with unknown reward distributions. The objective is to maximize cumulative rewards by balancing the exploration of uncertain arms and the exploitation of high-performing ones.

We formulate the topology adjustment as a CMAB problem, where each link $m_i$ in the model topology serves as an arm. In each outer loop $t$, the server selects an action $S_t$, which includes $K$ arms, where $K = d\langle m \rangle$. Selected arms $i \in S_t$ are activated, while others $i \notin S_t$ are deactivated, *i.e.,* $m_{t,i} = \mathbf{1}_{i \in S_t}$. After playing action, the new sparse weights $W_t = W_t^{agg} \odot m_t$ will be distributed to the clients for further training. The server then observes outcomes $X_t = (X_{t,1}, \cdots, X_{t,\langle m \rangle})$ for all arms, drawn from distributions with unknown expectation $\mu$. After that, the server obtains the unknown rewards $R_t = R(S_t, X_t)$. At the next round $t+1$, the previous information $\{X_\tau | 1 \le \tau \le t\}$ is the input to the adjustment algorithm to select the action $S_{t+1}$. Following previous work [26], we assume that the expected reward of an action $S_t$ only depends on $S_t$ and and the mean vector $\mu$, *i.e.,* there exists a function $r$ such that $\mathbb{E}[R_t] = \mathbb{E}_{X_t}[R(S_t, X_t)] = r(S_t, \mu)$. Assuming the number of adjustments is $T$, the goal of CMAB is to maximize the cumulative reward, *i.e.,* $\max \sum_{t=1}^{T} \mathbb{E}[R_t] = \sum_{t=1}^{T} r(S_t, \mu)$.

## 2.3 Challenges in Previous Federated Dynamic Pruning Methods

From the CMAB aspect, we can rigorously analyse the challenges in the previous methods. After taking the action $S_t$ to determine topology $m_t$, previous methods observe the delay outcomes $X_t$ at the next outer loop $t+1$, defining $X_{t,i} = 1$ for $i$-th arm with the top importance score; otherwise, $X_{t,i} = 0$. And the next action $S_{t+1}$ is selected **merely based on current outcomes** $X_t$, *i.e.,* $S_{t+1} = \{i | X_{t,i} = 1\}$. Thereby, they face three significant challenges:

**Greedy Adjustment**: Existing frameworks only observe the outcomes in the outer loop, ignoring the inner loop. Moreover, they discard all previous outcomes $\{X_\tau | 1 \le \tau \le t-1\}$, excluding information from unseen clients and previous data, leading to a myopic and greedy topology adjustment that is difficult to exploit from a global view.

**Unstable Topology**: Previous methods make a deterministic decision to select an action $S_t$ based on unreliable outcomes only, without considering the high variance of outcomes and the expectation of distribution $\mu$. Therefore, the selected action $S_t$ is hard to maximize the reward $r(S_t, \mu)$, leading to an unstable topology.

**Communication Inefficiency**: To compute the outcomes $X_t$ for all arms, existing frameworks use the magnitudes of aggregated weights and gradients as importance scores for active and inactive

---

[2]In this paper, $\langle \cdot \rangle$ is the cardinal number of the set.

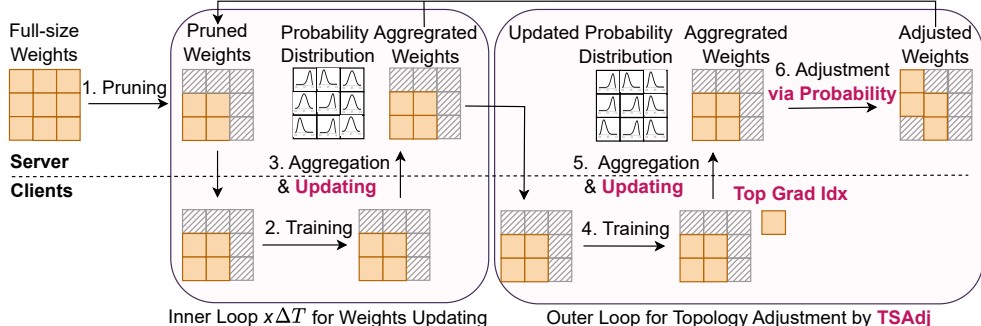

Figure 2: The overview of the FedRTS, which integrates TSAdj and utilizes the two-loop updating to develop a robust sparse model.

arms, respectively. As a result, clients must upload gradients for inactive weights, leading to a total communication cost comparable to that of a dense model, thus lacking communication efficiency. Although some approaches try to mitigate this by uploading only the top gradients [18, 19] or placing adjustments on the client side [1, 57], these greedy methods make the adjustment more unstable, leading to suboptimal performance.

## 3 Methodology

In this section, we present FedRTS, a novel federated pruning framework designed to develop robust model topologies, as illustrated in Fig. 2. We begin by detailing the TSAdj mechanism, which serves as the core component for searching robust sparse topologies. Next, we provide an overview of FedRTS that integrates TSAdj. Finally, we provide the theoretical regret upper bound of TSAdj.

### 3.1 Thompson Sampling-based Adjustment (TSAdj)

We find that the core source of the aforementioned challenges in existing methods lies in **deterministic decision strategies and myopic observations**. Therefore, we propose the Thompson Sampling-based Adjustment (TSAdj) mechanism, which includes a probabilistic decision strategy based on farsighted observations. Moreover, TSAdj effectively reduces communication overheads.

Compared to directly using myopic outcomes $X_t$ to make deterministic decisions, our proposed TSAdj maintains individual posterior probability distributions $P$ for each link in the model topology $m$ during training. The higher feedback a link $i$ has gained from previous observations, the closer the expectation of its probability $E[P_i]$ approaches 1, indicating a higher likelihood of being selected. In the outer loop for topology adjustment, the server samples random variables $\xi$, where $\xi_i \sim P_i$, and selects an action $S_t$ as follows:

$$S_t = \{i \in \text{Top}(\xi, K)\}, \tag{2}$$

where $\text{Top}(\xi, K)$ are the indices of top-$K$ elements in $\xi$. The action $S_t$ is determined by the posterior distributions $P$, rather than deterministically based on outcomes, thereby providing a more stable topology with lower variance [51] compared to existing approaches.

Unlike existing approaches that develop the outcomes using immediate and myopic data in the outer loop, TSAdj derives much more comprehensive outcomes $X_t$ from both the inner and outer loops. After performing action $S_t$, the outcome $X_t$ is observed in the end of loop $t$, and is formulated as a fusion of individual outcomes $X_t^n$ and aggregated outcomes $X_t^{agg}$:

$$X_t = \gamma X_t^{agg} + (1 - \gamma) \sum_{n \in C_t} p_n X_t^n, \tag{3}$$

where $\gamma$ is the trade-off ratio, $C_t$ is the set of clients that participate FL in loop $t$. This fusion mechanism mitigates biases that arise from unstable aggregated data.

We consider the semi-outcomes for $X_t^{agg}$ and $X_t^n$ in the inner loop, *i.e.,* only the arms $i \in S_t$ have the outcomes. We define a constant $\kappa < K$, which denotes the number of core links (arms) in the

model topology, and during the adjustment, we expect that $K - \kappa$ less important links are pruned, while $K - \kappa$ candidate links are reactivated. Following previous methods, we use the magnitude of weights to get the outcomes for active arms $i \in S_t$. Therefore, at the end of inner loop, we have

$$X_{t,i}^{agg} = h(i, |W_{t+1}^{agg}|, \kappa), \quad X_{t,i}^n = h(i, |W_{t+1}^n|, \kappa), \quad i \in S_t, \tag{4}$$

where $W_{t+1}^{agg}$ and $W_{t+1}^n$ represent the aggregated weights and client $n$'s weights, respectively. Here we use $t + 1$ instead of $t$ because the local training is finished. $h(\cdot)$ is a discriminative function defined as $h(i, x, \kappa) = \mathbf{1}_{i \in \text{Top}(x, \kappa)}$. In the inner loop, for arms $i \notin S_t$, both the aggregated outcomes $X_{t,i}^{agg}$ and the individual outcomes $X_{t,i}^n$ are set to $None$.

We consider the full outcomes of $X_t^{agg}$ and $X_t^n$ for outer loop. Beyond the outcomes of active arms in Eq. 4, we estimate the outcomes for inactive arms $i \notin S_t$ using the magnitudes of their gradients,

$$X_{t,i}^{agg} = 0.5, \quad X_{t,i}^n = h(i, |G_{t+1}^n|, K - \kappa), \quad i \notin S_t, \tag{5}$$

where the $G_{t+1}^n$ denotes the gradients of the weights $W_{t+1}^n$. Here, $K - \kappa$ is used to treat some inactivated arms as candidates for reactivation. We do not use the aggregated gradients $G_{t+1}^{agg}$ to compute $X_{t,i}^{agg}$ due to the high variance. Notably, to achieve Eq. 5, the clients only need to upload the Top indices of gradients magnitude, $\mathcal{I}_{t+1}^n = \text{Top}(|G_{t+1}^n|, K - \kappa)$, rather than the full gradients, significantly reducing the communication overhead.

To construct a posterior probability distribution, TSAdj updates the distribution $P$ based on outcomes $X_t$. Following the Thompson Sampling framework, each $P_i$ is modeled as a $Beta$ distribution with factors $\alpha_i, \beta_i$, representing the likelihood of selecting an arm into $S_t$: a higher $\alpha_i$ shifts $\mathbb{E}[P_i]$ towards 1, whereas a higher $\beta_i$ shifts it towards 0. The $(\alpha_i, \beta_i)$ are updated via Bayes' rule, leveraging its conjugacy properties and using the outcomes $X_{t,i}$ as follows:

$$(\alpha_i, \beta_i) \leftarrow (\alpha_i + \lambda X_{t,i}, \beta_i + \lambda(1 - X_{t,i})), \quad X_{t,i} \neq None, \tag{6}$$

where $\lambda$ is a scaling factor that determines the influence of outcomes. Over time, the distribution gradually converges towards the true distribution, aligning with the observed outcomes. The overflow of TSAdj is shown in Algorithm 1.

---

**Algorithm 1** TSAdj

---

**Input:** Participated clients $C_t$, uploaded sparse weights $\{W_{t+1}^n\}_{n \in C_t}$ and top gradients' indices $\{\mathcal{I}_{t+1}^n\}_{n \in C_t}$, trade-off ratio $\gamma$, scaling factor $\lambda$, global distribution factors $(\alpha, \beta)$, number of selected arms $K$, sparse model topology $m_t$.
**Output:** Updated sparse weights $W_{t+1}$ and topology $m_{t+1}$
$W_{t+1}^{agg} \leftarrow \sum_{n \in C_t} p_n W_{t+1}^n$
$X_t^{agg}, \{X_t^n\}_{n \in C_t} \leftarrow$ calculate with $W_{t+1}^{agg}, \{W_{t+1}^n\}_{n \in C_t}$ and $\{\mathcal{I}_{t+1}^n\}_{n \in C_t}$ by Eq. 4 and Eq. 5
$X_t \leftarrow \gamma X_t^{agg} + (1 - \gamma) \sum_{n \in C_t} p_n X_t^n$
$(\alpha_i, \beta_i) \leftarrow (\alpha_i + \lambda X_{t,i}, \beta_i + \lambda(1 - X_{t,i})), \quad$ for $i = 1, 2, \dots, \langle X_t \rangle$ and $X_{t,i} \neq None$
$S_{t+1} \leftarrow \{i | \xi_i \in \text{Top}(\xi, K)\}, \quad \xi = \{\xi_i | \xi_i \sim Beta(\alpha_i, \beta_i)\}$ // Select new action
$m_{t+1} \leftarrow \mathbf{1}_{\{i \in S_{t+1}\}}$ // Adjust topology based on selected action
$W_{t+1} \leftarrow W_{t+1}^{agg} \odot m_{t+1}$

---

In summary, TSAdj leverages the posterior probability distributions $P$ to probabilistically guide topology adjustments and uses comprehensive outcomes $X_t$ to update the distributions, which mitigates instability caused by myopic adjustments, enabling smoother convergence and developing a more reliable topology. Moreover, TSAdj requires the clients to upload only the most critical gradient information, thereby reducing the communication overhead.

### 3.2 Proposed FedRTS framework

FedRTS is a novel federated pruning framework designed to address the limitations of existing dynamic sparse training methods by integrating the Thompson Sampling-based Adjustment (TSAdj) mechanism, as illustrated in Fig. 2. The framework begins with an initialized model weight and sparse topology sampled from Beta distributions $P$. FedRTS employs a two-loop training process to iteratively update model weights and refine the sparse topology.

In the inner loop, the model topology remains fixed while the server updates the model weights and distribution $P$ based on semi-outcomes. In the outer loop, the server collects the full outcomes by requiring clients to upload the top gradient indices of inactivated weights. Using this information, the server applies TSAdj to make robust topology adjustments, and the updated topology is then distributed to clients. FedRTS continues this iterative process until model convergence. The details of FedRTS are shown in Algorithm 2 and Sec. B.2.

By integrating TSAdj, FedRTS introduces a novel approach to sparse topology adjustment. Leveraging Thompson Sampling, it mitigates the instability caused by unstable aggressive topology, ensuring smoother convergence through probabilistically guided adjustments based on historical information. This adaptive adjustment strategy enhances the robustness of the sparse topology and improves overall model performance in FL tasks.

### 3.3 Overhead Analysis

The overhead of TSAdj arises from maintaining and sampling the Beta distribution on the server. In each round, the server performs the following operations with $\langle C_t \rangle$ sampled clients: weight averaging with a time complexity of $\mathcal{O}(\langle C_t \rangle \langle W \rangle)$, computing the outcome using Quicksort with an expected time complexity of $\mathcal{O}(\langle C_t \rangle \langle W \rangle)$, element-wise Beta distribution updates in $\mathcal{O}(\langle W \rangle)$, and arms selection in $\mathcal{O}(\langle W \rangle)$. The overall complexity thus simplifies to $\mathcal{O}(\langle C_t \rangle \langle W \rangle)$. Compared to the baseline complexities of FedMef, FedTiny, and PruneFL, which are also $\mathcal{O}(\langle C_t \rangle \langle W \rangle)$, TSAdj achieves the same computational order. This demonstrates that TSAdj introduces no significant additional computational burden on the server, even with large models.

**Critically, TSAdj introduces no additional computations on client devices compared to the federated pruning baseline, as its overhead is entirely server-side.** In cross-device federated learning, the server is typically assumed to possess substantial computational resources; therefore, TSAdj only introduces a slight overhead on the system.

### 3.4 Theoretical Analysis

For simplifications, we omit the inner loop of FedRTS to isolate the TSAdj mechanism and consider only semi-outcomes, *i.e.,* ignore the outcomes $X_{t,i}$ that $i \notin S_t$. We adopt the following assumptions:

**Assumption 3.1.** *(L-continuity)* $\exists L \in \mathbb{R}, \forall S, \mu, \mu'$ *satisfies* $|r(S, \mu) - r(S, \mu')| \leq L \|\mu - \mu'\|_1$.

**Assumption 3.2.** *(Independent importance score) The importance scores used for pruning are treated as mutually independent.*

**Assumption 3.3.** *(Mean-field Approximation) when* $\{x_i\}$ *are mutually independent, the discriminative function* $h(\cdot)$ *admits a mean-field approximation:* $h(i, x, \kappa) = \mathbf{1}_{i \in \{x, \kappa\}} \approx \mathbf{1}_{x_i \geq \sigma}$, *where the threshold* $\sigma$ *is determined self-consistently via mean-field decoupling (independent of individual* $x_i$) *and satisfies* $\langle x \rangle - \sum_i^{\langle x \rangle} \mathrm{CDF}_i(\sigma) = \kappa$, *where* $\mathrm{CDF}_i$ *is the cumulative distribution function of* $x_i$.

Assumption 3.1 is standard in bandit optimization work [26]. Assumption 3.2 is implicitly or explicitly adopted in pruning methods [46, 12, 52]. Assumption 3.3 holds for large scale $\langle x \rangle$ (>100k) [53, 54, 35].

While weight magnitudes are not strictly theoretically independent when used as importance scores, both empirical and theoretical evidence supports simplification in Assumption 3.2: (1) Recent work demonstrates that trained weights in large networks exhibit asymptotic independence [54, 53], and (2) this assumption is well-established in the pruning work [12, 46], where weight dependencies are typically disregarded during selection. While other importance scores (*e.g.,* Fisher information [52]) might better satisfy independence, they are computationally prohibitive in FL. Thus, we follow prior works [24, 19] to apply weight magnitude pruning, acknowledging a minor discrepancy between theory and practice. Under assumptions 3.2 and 3.3, outcomes $X_{t,i}$ simplify to:

$$X_{t,i} = \gamma \mathbf{1}_{|W_{t,i}| \geq \sigma} + (1 - \gamma) \sum_n^{\mathcal{N}} p_n \mathbf{1}_{|W_{t,i}^n| \geq \sigma}, \quad i \in S_t, \tag{7}$$

which is mutually independent.

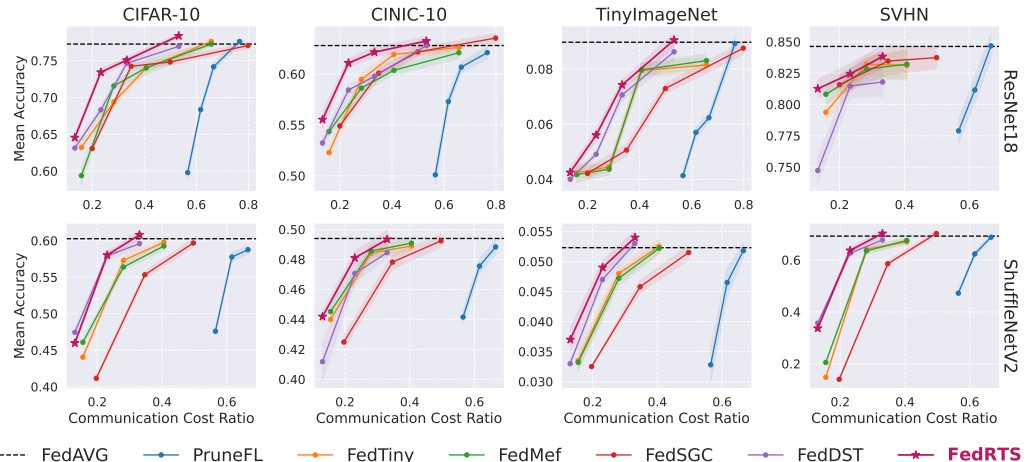

Figure 3: Testing accuracy of FedRTS and different federated pruning baselines on the four CV datasets with different densities, where the ratio is communication cost relative to dense FedAVG.

To analyze the worst-case performance of TSAdj, we define the cumulative regret $Reg(T)$ as:

$$Reg(T) = \mathbb{E}\left[\sum_{t=1}^{T} \Delta_{S_t}\right], \tag{8}$$

where $\Delta_{S_t} = \max\{r(S^*, \mu) - r(S_t, \mu), 0\}$ and $S^*$ is the greedy optimal action based on $\mu$, which is obtained by Algorithm 3. The maximum reward gap is set as $\Delta_{\max} = \max_S \Delta_S$. The maximum reward gap of actions containing arm $s$ is $\Delta_s^{\max} = \max_{S:s\in S} \Delta_S$. We define $S^*$ and $S_t$ are sequences, i.e., $S^* = (s_1^*, \ldots, s_K^*)$, $S_t = (s_{t,1}, \ldots, s_{t,K})$, and $s_k^* \in S^*$ is the $k$-th element added to $S^*$. We denote $S_k^* = (s_1^*, \ldots, s_k^*)$ and $S_{t,k} = (s_{t,1}, \ldots, s_{t,k})$. For any arm $s$, define the marginal reward gap $\Delta_{s,k} = r(S_k^*, \mu) - r(S_{k-1}^* \cup \{s\}, \mu)$. Inspired by previous work [5, 26], we provide a regret upper bound for TSAdj.

**Theorem 3.4.** *(Upper Bound) Under assumptions 3.2, 3.3 and 3.1 and with outcomes $X_t$ defined in Eq. 7, the regret $Reg(T)$ of TSAdj can be upper bounded by:*

$$Reg(T) \leq \sum_{s\neq s_1^*} \max_{k:s\notin S_k^*} \frac{6\Delta_s^{\max}\log T}{(\Delta_{s,k} - 2LK\epsilon)^2} + \left(2K + 4\langle W\rangle + \frac{KU + 8K\epsilon^4}{\epsilon^6}\right)\Delta_{max}$$

$$= O\left(\sum_{s\neq s_1^*} \max_{k:s\notin S_k^*} \frac{\Delta_{max}L^2\log T}{\Delta_{s,k}^2} + \Delta_{max}\right). \tag{9}$$

*for any $\epsilon$ such that $\forall s \neq s_1^*$ and $s \notin S_k^*, \Delta_{s,k} > 2LK\epsilon$, where $U$ is a universal constant.*

The proof is provided in Sec. C.2. **Crucially, our theorem hold for any pruning method satisfying Assumption 3.2 and 3.3**, including future advancements in efficient independent-scoring techniques.

## 4 Evaluation

To validate the efficacy of FedRTS, we conduct comprehensive experiments in this section. Notably, we evaluate each experiment five times, reporting the mean results along with the standard errors. In the figures, the shaded area represents the standard errors. In the table, the best results are highlighted in **purple** color, while the second-best results are marked in **blue** color.

### 4.1 Experiment Setup

We provide an overview of the experimental setup in this section, with additional details in Sec. F.

| Target Density | Method | PPL ↓ | Avg. Acc ↑ | Comm. Cost ↓ |
|---|---|---|---|---|
| 1 | FedAVG | 20.56 | 0.4387 | 260.41MB |
| 50% | FedDST | **20.10** | 0.4261 | **138.30 MB** |
| | FedSGC | 26.13 | 0.4110 | 207.45 MB |
| | FedMef | 20.61 | 0.4352 | 165.96 MB |
| | **FedRTS** | **18.54** | **0.4422** | **138.84 MB** |
| 30% | FedDST | 24.46 | 0.4263 | **86.26 MB** |
| | FedSGC | 32.39 | 0.3939 | 129.39 MB |
| | FedMef | 21.00 | 0.4304 | 103.51 MB |
| | **FedRTS** | **18.56** | **0.4405** | **86.44 MB** |
| 20% | FedDST | 26.49 | 0.4219 | **60.22 MB** |
| | FedSGC | 43.00 | 0.3550 | 90.33 MB |
| | FedMef | 21.53 | 0.4293 | 72.26 MB |
| | **FedRTS** | **19.93** | **0.4333** | **60.34 MB** |

Table 1: Performance comparison on TinyStories with GPT-2-32M.

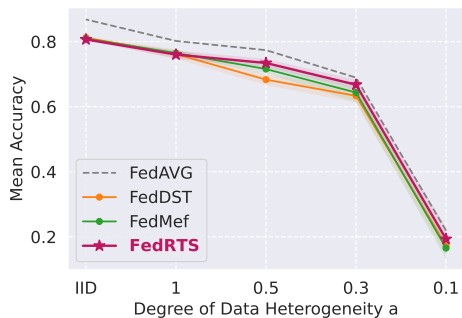

Figure 4: Accuracy on CIFAR-10 under various degrees of data heterogeneity.

**Datasets and Models.** Following previous work [18, 19, 1, 24] on federated dynamic pruning, we conduct experiments on CV tasks using two lightweight models, ResNet18 [13] and ShuffleNetV2 [66], across four well-known image classification datasets: CIFAR-10 [27], CINIC-10 [7], TinyImageNet [8] and SVHN [43]. For NLP tasks, we use the GPT-2-32M model on the TinyStories dataset [10], a language understanding benchmark designed for small language models.

**Federated Learning Settings.** Following the settings used in previous frameworks [18, 19, 1], we conduct $T_{max} = 500$ communication rounds with 5 local training epochs. Batch sizes are set to 64 for CV tasks and 16 for NLP tasks. In each round, 10 clients are randomly selected from a total of $\mathcal{N} = 100$ clients. To generate data heterogeneity on the client, we utilize the Dirichlet distribution with factor $a$. The types of heterogeneity for CV and NLP tasks are different. CV tasks apply label imbalance with $a = 0.5$ by default. Unlabeled NLP tasks set data quantity imbalance, with $a = 5$ by default. We perform an outer loop for topology adjustment every $\Delta T = 10$ inner loops until the total number of loops $t$ reaches $T_{end} = 300$. The optimizer used is SGD, with a learning rate of $\eta = 0.01$.

**Baseline Settings.** We select the following SOTA federated pruning frameworks as baselines: PruneFL [24], FedTiny [18], FedDST [1], FedSGC [57], and FedMef [19]. These methods incorporate dynamic pruning in FL and achieve SOTA performance across various settings, as detailed in Sec. F.2.

**Implementation Details.** Following previous work [18, 19], we exclude bias terms, normalization layers, and the output layer from pruning while ensuring the overall density $d \leq d'$. Layer-wise sparsity follows the Erdos-Renyi-Kernel distribution [11]. We set the scaling factor $\lambda = 10$ to match the number of selected clients per loop, ensuring that the averaged information maintains individual-level influence. The trade-off ratio is set as $\gamma = 0.5$ to assign equal importance to the individual and aggregated model. We set the layer-wise $\kappa^l = K^l - \frac{\alpha_{adj}}{2}(1 + \cos(\frac{t\pi}{T_{end}}))K^l$, where $K^l$ is the number of active weights at $l$-th layer and $\alpha_{adj}$ is 0.4 by default, is adopted directly from prior works [19, 11]. Training FLOPs and communication costs are calculated as described in Sec. F.4.

### 4.2 Performance Evaluation

To show the effectiveness of FedRTS, we conduct experiments on CV and NLP tasks under various target density levels. Broader experiments including efficiency analysis are in Sec. E.2 and E.3.

**Computer Vision Tasks.** In the CV tasks, we conduct experiments using the ResNet18 and ShuffleNetV2 models, testing four target density levels: 50%, 30%, 20%, and 10% on CIFAR-10, CINIC-10, TinyImageNet, and SVHN datasets, reporting the accuracy and communication costs ratio per outer loop. In some cases, FedRTS achieves similar accuracy to FedAVG at the 30% density ratio; therefore, results for the 50% density ratio are omitted. As shown in Fig. 3, FedRTS consistently outperforms all baseline methods in most scenarios in terms of both accuracy and communication efficiency. For instance, in the experiment with ResNet18 in CIFAR-10, FedRTS achieves a 5.1% improvement in accuracy over the best baseline methods while maintaining a similar communication cost. These gains are attributed to the robust adjustment, stable topology, and communication efficiency provided by TSAdj in FedRTS.

An evident trend is the superior accuracy achieved by ResNet18 compared to ShuffleNetV2, due to ResNet18's larger parameter space and higher representational capacity. Furthermore, FedDST and

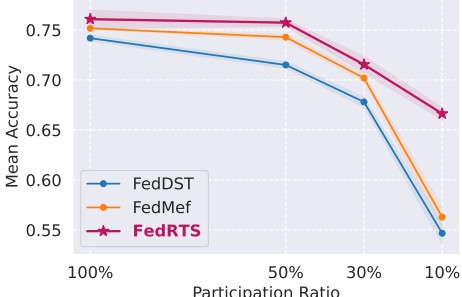
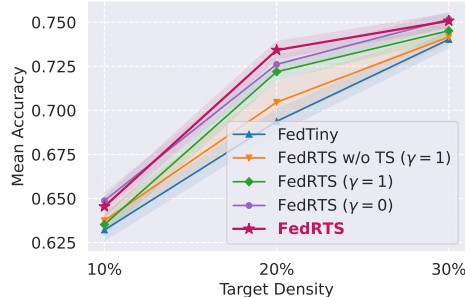

Figure 5: Accuracy on CIFAR-10 with various participation ratios of clients.

Figure 6: Accuracy on CIFAR-10 for FedRTS and various variants.

FedMef consistently outperform other baseline methods. Based on these empirical observations, we select ResNet18 as the default CV model and identify FedDST and FedMef as the primary reference for subsequent experiments.

**Natural Language Processing Tasks.** To evaluate the effectiveness of FedRTS across different domains, we conduct experiments on NLP tasks using the TinyStories dataset with the GPT-2-32M model. We test three target density levels: 50%, 30%, and 20%. Perplexity (PPL), which assesses model performance by considering the entire probability distribution in generated text, serves as the primary evaluation metric. As shown in Tab. 1, FedRTS outperforms all baselines, demonstrating its superiority in the NLP domain. Notably, FedRTS achieves better performance in perplexity than in accuracy-based evaluations, even surpassing full-size FedAVG at low-density levels. This improvement may stem from its probabilistic adjustment strategy, which enhances flexibility and adaptability during optimization. Unlike deterministic approaches, probabilistic adjustment enables FedRTS to prioritize critical parameters effectively, improving its ability to adapt to complex probability distributions. This is particularly beneficial for NLP tasks, where capturing intricate text patterns is essential, leading to lower perplexity and stronger performance in low-resource settings.

## 4.3 Robustness Analysis

To demonstrate the robustness of FedRTS in the face of heterogeneous data distributions and partial client participation, we conduct experiments with different degrees of data heterogeneity and client availability in this section. Further experiments on the impact of the hyperparameters $\gamma$, $\lambda$, $\kappa$, and $\Delta T$ are provided in Sec. E.1.

**The Impact of Data Heterogeneity.** Data heterogeneity, or non-independent and identically distributed (non-IID) data, is a critical factor affecting performance in FL. To demonstrate the robustness of FedRTS, we conduct experiments with varying degrees of data heterogeneity on the CIFAR-10 and TinyStories datasets. By adjusting the Dirichlet distribution factor $a$, we control the degree of data heterogeneity, where a lower $a$ indicates higher heterogeneity. As shown in Fig. 4 and Fig. 14, FedRTS outperforms all baselines, particularly in highly heterogeneous settings, demonstrating its effectiveness in handling non-IID data. This improvement is primarily attributed to the stable topology produced by TSAdj in FedRTS.

**The Impact of Client Availability.** Client availability is another significant challenge in FL. To evaluate the robustness of FedRTS under different levels of client availability, we conduct experiments with participation ratios of 100%, 50%, 30%, 10% using a total of 10 clients. To mitigate slow convergence at lower participation levels, we adopt a high learning rate of 1 with SGD, as Adam exhibited instability with NaN values in our experiments. As shown in Fig. 5, FedRTS consistently outperforms all baselines, particularly at low participation ratios, demonstrating its adaptability and robustness in limited-client scenarios.

## 4.4 Ablation Study

To analyze the individual contributions of FedRTS in addressing greedy and unstable topology adjustment problems, we conduct an ablation study on the CIFAR-10 dataset with various target

density levels. We consider the benefits of FedRTS to come from three aspects: probabilistic decision, farsighted information, and stability from the fusion mechanism in Eq. 3. To evaluate these aspects independently, we consider the following variants of FedRTS: (1) *FedRTS ($\gamma = 0$)*: This variant does not use the aggregated information to update the probability distribution. (2) *FedRTS ($\gamma = 1$)*: This variant only uses the aggregated information to update the probability distribution, which disables the fusion mechanism to reduce stability. (3) *FedRTS without TS ($\gamma = 1$)*: This variant fuses historical information via a momentum mechanism similar to FedAdam [49] and uses deterministic adjustment to obtain the new topology, disabling both probabilistic decision and fusion mechanism. (4) *FedTiny*: This serves as a baseline to evaluate the overall effectiveness of FedRTS, disabling all three aspects.

The results in Fig. 6 demonstrate the effectiveness of FedRTS. First, the performance of *FedRTS* and *FedRTS ($\gamma = 0$)* is similar but better than *FedRTS ($\gamma = 1$)*, demonstrating that aggregated information suffers from instability. A fusion mechanism including individual information can make the adjusted topology more stable and achieve better performance. Second, without deterministic adjustment, *FedRTS w/o TS ($\gamma = 1$)* performs much worse than *FedRTS ($\gamma = 1$)*, indicating the benefits of probabilistic adjustment. Third, *FedRTS w/o TS ($\gamma = 1$)* is better than *FedTiny*, showing the effectiveness of farsighted information. Thus, the ablation study demonstrates FedRTS's effectiveness in terms of probabilistic decision, farsighted information, and stability from the fusion mechanism.

## 5    Limitation

While FedRTS with TSAdj effectively addresses key challenges: greedy adjustments, unstable topologies, and communication inefficiency in federated dynamic pruning, our approach shares a common limitation with prior magnitude-based pruning methods [12, 46]. The theoretical analysis relies on Assumption 3.2, which presumes independence among importance scores. In practice, weight magnitudes exhibit only asymptotic independence [53], creating a minor but acknowledged gap between theory and empirical results.

We considered alternative importance metrics like Fisher information [52] that might better satisfy the independence requirement. However, their computational complexity makes them impractical for federated settings with resource-constrained clients. Following established practice in the pruning literature, we therefore adopt magnitude-based pruning while transparently acknowledging this theoretical-empirical discrepancy, a compromise that reflects the inherent trade-offs in practical federated learning systems.

While the mean-field approximation in Assumption 3.3 provides theoretical grounding for TSAdj, its validity weakens for extremely small models (<100K parameters) where finite-size effects become significant [53]. However, at such small scales, the benefits of pruning become marginal, as these models can typically be trained directly on-device without compression. Thus, this limitation has minimal practical impact on FedRTS.

The FedRTS also presents the following two limitations. Fairness and bias: The method dynamically adjusts model structure per global feedback. In highly heterogeneous settings, this could unintentionally bias the model toward data-rich or majority-class clients. Security/privacy considerations: While the method reduces communication cost, sharing top gradient indices might still leak sensitive information.

## 6    Conclusion

In this paper, we analyze federated pruning from the perspective of combinatorial multi-armed bandits and identify key limitations in existing methods, including greedy adjustments, unstable topologies, and communication inefficiencies. To address these challenges, we propose Federated Robust Pruning via Combinatorial Thompson Sampling (FedRTS), a novel framework for developing robust sparse models. The core component of FedRTS, the Thompson Sampling-based Adjustment (TSAdj) module, enhances robustness and performance by making probabilistic decisions based on stable, farsighted information rather than deterministic decisions driven by unstable, myopic observations. We provide a theoretical regret upper bound for TSAdj and conduct extensive experiments, demonstrating that FedRTS achieves state-of-the-art performance in both computer vision and natural language processing tasks.

## Acknowledgments and Disclosure of Funding

This paper is partially supported by Hong Kong Research Grants Council (RGC) grant C1042-23GF and grant #11203523 and Hong Kong Innovation and Technology Fund (ITF) grant MHP/061/23 and grant MHP/034/22.

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

# Appendices Contents

# A Related Work

## A.1 Neural Network Pruning

Neural network pruning [41, 28, 21] is a widely used technique for reducing the computational and memory footprint of neural networks by eliminating redundant parameters. Pruning methods can be categorized into structured pruning [36, 14, 32, 39], which removes entire structures like filters or layers, and unstructured pruning [12, 21], which targets individual weights. While structured pruning simplifies model deployment by maintaining regular architectures, it often suffers from significant performance degradation at higher sparsity levels (*e.g.,* beyond 10% [39]), limiting its compression potential. In contrast, unstructured pruning can achieve sparsity levels exceeding 50% with minimal accuracy loss, making it a more flexible approach. For this reason, our discussion focuses primarily on unstructured pruning.

Traditional pruning pipelines follow a train-prune-retrain paradigm: a dense model is first trained, less important weights are pruned based on some criterion (*e.g.*, magnitude [21, 12], gradients [41, 38], or Fisher information [52]), and the model is then fine-tuned to recover performance. However, this process is computationally expensive, as it requires training a dense model before pruning.

To avoid this overhead, recent work explores dynamic sparse training (DST) [37, 9, 11], where sparse topologies are optimized during training without ever densifying the model. While promising, existing DST methods rely on deterministic pruning-and-regrowth strategies, which can lead to high variance in model performance and inefficient exploration of sparse topologies, especially in federated learning scenarios with heterogeneous data and partial client participation. To mitigate these challenges, we propose TSAdj, a probabilistic adjustment mechanism that enables more stable and robust sparse topology evolution.

## A.2 Federated Pruning

Federated Pruning is a technique that applies neural network pruning within the Federated Learning (FL) process. To avoid the requirements of dense training in traditional pruning methods, some previous methods determine the sparse model topology before federated training and do not adjust it during the FL process, such as SNIP [29] and Synflow [56]. This approach of pruning at initialization may overlook crucial data information, potentially leading to suboptimal model structures.

Dynamic sparse training methods, such as those employed by PruneFL [24], adjust the model during FL rounds. However, these methods require the entire model gradient to be sent to the server to ensure final performance, resulting in high communication costs. FedDST [1] and FedSGC [57] alleviate this issue by applying dynamic sparse training on clients. FedTiny [18] and FedMef [19] require clients only to upload the Top gradients. Despite these advancements, these methods adjust the model topology based on limited information from the currently selected clients, leading to issues of greedy adjustment and unstable topology. FedRTS addresses these challenges by applying TSAdj, significantly mitigating the problems associated with limited client information and enhancing the robustness of model adjustments by leveraging probabilistic distributions.

## A.3 Combinatorial Multi-Arm Bandit

The Multi-Armed Bandit (MAB) problem [55] is a fundamental framework for decision-making under uncertainty, where an agent repeatedly selects among multiple actions ("arms") with initially unknown rewards to maximize cumulative gains. The Combinatorial Multi-Armed Bandit (CMAB) [5] extends this classic problem to scenarios where rewards depend on combinations of selected arms rather than individual choices.

Two primary approaches have emerged for solving CMAB problems: The first is the Combinatorial Upper Confidence Bound (CUCB) [5], which uses optimistic reward estimates to balance exploration and exploitation. The second is Combinatorial Thompson Sampling (CTS) [6, 59], which employs probabilistic sampling from posterior distributions to dynamically adjust its strategy.

While CUCB's deterministic nature makes it theoretically appealing for analysis, its inherent optimism [5] can lead to persistently selecting clearly suboptimal arm combinations. This behavior prevents stable convergence, making CUCB less suitable in dynamic pruning where consistent per-

formance is crucial, as will be discussed in detail in Sec. E.3.1. For this reason, we built our TSAdj method on the more adaptive CTS framework.

# B Algorithm

## B.1 TSAdj

In each loop $t$, the aggregated weights $W_{t+1}^{agg}$ will be calculated in each loop when the server receives the weights $W_{t+1}^n$ from each client $n \in C_t$, and TSAdj is invoked to adjust the sparse topology $m_t$ only in the outer loop. The outcomes $X_{t,i}$ for link $m_i$ are computed as described in Equation 3, which balances global and client-specific contributions. Specifically, outcomes $X_{t,i}^{agg}$ and $X_{t,i}^n$ are calculated based on Equations 4 on the server, using the aggregated weights $W_{t+1}$ and the uploaded weights $W_{t+1}^n$, while $X_{t,i}^n$ will be updated in the outer loop using the indices of the top $K - \kappa$ gradients $\mathcal{I}_{t+1}^n$ from each client $n$ based on Equation 5. Outcomes $X_{t,i}^{agg}$ and $X_{t,i}^n$ are combined to compute the final outcome $X_{t,i}$ for link $m_i$ in Equation 3, where $\gamma \in [0, 1]$ controls the trade-off between global and client-specific contributions, and $p_n$ reflects the relative weight of client $n$ based on the size of its local dataset $D_n$. A larger $\gamma$ emphasizes client-specific observations, while a smaller $\gamma$ prioritizes global contributions.

The posterior probability distribution $P$ for each link $m_i$ in the topology $m$ is modeled as a beta distribution parameterized by $(\alpha_i, \beta_i)$. At initialization, $(\alpha_i, \beta_i)$ are set to small positive values (*e.g.,* $\alpha_i = \beta_i = 1$) to represent a uniform prior, ensuring that all links have an equal probability of activation at the outset. The beta parameters $(\alpha_i, \beta_i)$ for each link $m_i$ are then updated incorporating with the computed outcomes $X_{t,i}$, via Equation 6, where $\lambda$ controls the trade-off between exploration and exploitation by adjusting the impact of the current outcomes on the beta distribution's update. A larger $\lambda$ increases the influence of the current loop's outcomes $X_{t,i}$, accelerating the concentration of the Beta distribution towards a certain probability. Conversely, a smaller $\lambda$ preserves uncertainty to encourage exploration. Over time, as $\alpha_i$ and $\beta_i$ grow under the influence of $\lambda$ and $X_{t,i}$, the Beta distribution becomes sharper and more concentrated, reflecting an increasingly certain probability of selecting $m_i$ as an active link in the topology $m$.

In each loop $t$, the topology adjustment dynamically activates or deactivates links based on a probabilistic action $S_t$ in Equation 2, where $\xi$ is sampled from the updated posterior distribution $P$. For each link $m_i$, the probability $\xi_i$ is independently sampled from its corresponding Beta distribution $P_i$. Links are activated if their probabilities $\xi_i$ rank among the top $K$ in magnitude. The TSAdj mechanism outputs the updated sparse topology $m_{t+1}$ and the weights $W_{t+1} \odot m_{t+1}$, where $\odot$ denotes element-wise multiplication. The updated mask $m_{t+1}$ represents the optimized topology for the next training loop $t + 1$.

## B.2 FedRTS

The framework of FedRTS begins by initializing the global model weights $W_0$ and a sparse mask $m_0$ with a predefined sparsity level $d'$. The sparse mask $m_0$ determines the active parameters in the initial global model. Additionally, hyperparameters such as the total number of training loops $T_{end}$ and the adjustment interval $\Delta T$ are set. These parameters control the frequency of topology adjustments and the overall training duration.

The training process is conducted over $T_{max}$ loops. At the beginning of each training loop $t$, the server employs a random sampling mechanism to select a subset of clients $C_t$ from the total $\mathcal{N}$ available clients for participation in the training process. This sampling process is conducted uniformly at random, ensuring that each client, indexed by $n$ from $1, \ldots, \mathcal{N}$, has an equal probability of being chosen. This randomized partial participation mechanism not only avoids bias in client selection but also ensures scalability and reduces communication overhead, making it well-suited for large-scale distributed systems.

The server then broadcasts the global sparse weights $W_t$ and the corresponding mask $m_t$ to all participating clients $C_t$. Each selected client $n \in C_t$ trains the received model $(W_t, m_t)$ on its local dataset $D_n$ for a fixed number of loops, producing intermediate outputs. Specifically, gradients $G_{t+1}^n$ will be computed only during outer loops for topology adjustment. The client identifies the gradients with the top $K - \kappa$ magnitudes based on the gradients $G_{t+1}^n$ and upload its corresponding

---

**Algorithm 2** FedRTS

---

**Input:** Initial model $W_0$, target sparsity $d'$, learning rate $\eta$, maximum of training loops $T_{end}$, maximum of all loops $T_{max}$, adjustment interval $\Delta T$, training datasets $\{D_n\}_{n \in \mathcal{N}}$, the number of core links $\kappa$,
**Output:** Final sparse model $W_{T_{max}}, m_{T_{max}}$

Randomly initialize $m_0$ with sparsity $d'$,
**for** each loop $t = 0, 1, 2, \ldots, T_{max} - 1$ **do**
    // The server does
    Sample a random subset of clients $C_t$
    Broadcast server sparse model $(W_t, m_t)$ to all clients $n \in C_t$

    **for** each client $n \in C_t$ **do**
        // The client $n$ does
        $W_t^n, m_t^n \leftarrow W_t, m_t$ // Fetch global model
        $W_{t+1}^n \leftarrow W_t^n - \eta \nabla f_n(W_t^n, m_t, D_n) \odot m_t$
        Transmit $W_{t+1}^n$ to server
        **if** $t \bmod \Delta T = 0$ and $t < T_{end}$ **then**
            Sample batch data $\mathcal{B}_n \sim D_n$
            $G_{t+1}^n \leftarrow \nabla f_n(W_{t+1}^n, m_t, \mathcal{B}_n) \odot m_t$
            $\mathcal{I}_{t+1}^n \leftarrow \{i | i \in \text{Top}(|G_{t+1}^n|, K - \kappa)\}$ // Top $K - \kappa$ magnitudes' indices
            Transmit $\mathcal{I}_{t+1}^n$ to server
        **end if**
    **end for**

    // The server does
    **if** $t \bmod \Delta T = 0$ and $t < T_{end}$ **then**
        $W_{t+1}^{agg}, m_{t+1} \leftarrow \text{TSAdj}(\{W_{t+1}^n\}_{n \in C_t}, \{\mathcal{I}_{t+1}^n\}_{n \in C_t})$ // TSAdj in Algorithm. 1.
    **else**
        // Update beta distribution based on weights' outcomes
        $W_{t+1}^{agg} \leftarrow \sum_{n \in C_t} p_n W_{t+1}^n$
        $m_{t+1} \leftarrow m_t$   // inner loop do not update topology
        $X_t^{agg}, \{X_t^n\}_{n \in C_t} \leftarrow$ calculate with $W_{t+1}^{agg}$ and $\{W_{t+1}^n\}_{n \in C_t}$ by Eq. 4
        $X_t \leftarrow \gamma X_t^{agg} + (1 - \gamma) \sum_{n \in C_{t+1}} p_n X_t^n$
        **for** each links $i = 1, 2, \ldots, \langle X_t \rangle$ **do**
            $(\alpha_i, \beta_i) \leftarrow (\alpha_i + \lambda X_{t,i}, \beta_i + \lambda(1 - X_{t,i})), X_{t,i} \neq None$
        **end for**
    **end if**
**end for**

---

indices $\mathcal{I}_{t+1}^n = \{i | i \in \text{Top}(|G_{t+1}^n|, K - \kappa)\}$, representing the most promising links for the potential reactivation, rather than the complete gradients to the server, further reducing communication overhead. In contrast, weights $W_{t+1}^n$ are computed and uploaded to the server during both inner and outer loops to facilitate global aggregation. Unlike gradients, which are selectively uploaded, the complete set of weights is transmitted to ensure comprehensive integration into the global model.

Upon receiving the locally updated gradient indices $\mathcal{I}_{t+1}^n$ in the outer loop and weights $W_{t+1}^n$ from the participating clients, the server aggregates the weights to update global model weights $W_{t+1}^n$ using a weighted average formula. Mathematically, the aggregated weights $W_{t+1}^{agg}$ are computed as: $W_{t+1}^{agg} = \sum_{n \in C_t} p_n W_{t+1}^n$. If the current loop $t$ is an outer loop (i.e., $t \bmod \Delta T = 0$), the server invokes the TSAdj mechanism to adjust the sparse mask $m_t$. The updated mask $m_{t+1}$ is then applied to the global model.

After completing $T_{end}$ training loops, the server outputs the final globally aggregated weights $W_{T_{max}}$ and an optimized mask $m_{T_{max}}$, both derived from the iterative training involving all participating clients. The weights $W_{T_{max}}$ represent the learned parameters that effectively accommodate heterogeneous data distributions, while the topology $m_{T_{max}}$ preserves the robust and reliable structure of the model.

# C  Theoretical Details

## C.1  Greedy Optimal Action

---
**Algorithm 3** Greedy Optimal Action

---
  **Input:** The mean vector $\mu$ and action size $K$
  **Return:** The greedy optimal action $S^*$
  Initialize: $S^* = \emptyset$
  **for** $k \in 1, \ldots, K$ **do**
    $S^* = S^* \cup \{\text{argmax}_{i \neq S^*} r(S^* \cup \{i\}, \mu)\}$
  **end for**

---

## C.2  Proof of Theorem 3.4

The cumulative regret $Reg(T)$ is defined as:

$$Reg(T) = \mathbb{E}\left[\sum_{t=1}^{T} \Delta_{S_t}\right], \tag{10}$$

where $\Delta_{S_t} = \max\{r(S^*, \mu) - r(S_t, \mu), 0\}$ and $S^*$ is the greedy optimal action based on $\mu$, which is obtained by Alg. 3. The maximum reward gap is set as $\Delta_{\max} = \max_S \Delta_S$. The maximum reward gap of actions containing arm $s$ is $\Delta_s^{\max} = \max_{S:s \in S} \Delta_S$. We define $S^*$ and $S_t$ are sequences, *i.e.*, $S^* = (s_1^*, \ldots, s_K^*)$, $S_t = (s_{t,1}, \ldots, s_{t,K})$, and $s_k^* \in S^*$ is the $k$-th element added to $S^*$. We denote $S_k^* = (s_1^*, \ldots, s_k^*)$ and $S_{t,k} = (s_{t,1}, \ldots, s_{t,k})$. For any arm $s$, define the marginal reward gap $\Delta_{s,k} = r(S_k^*, \mu) - r(S_{k-1}^* \cup \{s\}, \mu)$.

**Theorem.** *(Upper Bound) Under assumptions 3.2, 3.3 and 3.1 and with outcomes defined in Eq. 7, the regret $Reg(T)$ of TSAdj can be upper bounded by:*

$$
\begin{aligned}
Reg(T) &\leq \sum_{s \neq s_1^*} \max_{k:s \notin S_k^*} \frac{6\Delta_s^{\max} L^2 \log T}{(\Delta_{s,k} - 2LK\epsilon)^2} + \left(2K + 4\langle W \rangle + \frac{KU + 8K\epsilon^4}{\epsilon^6}\right) \Delta_{max} \\
&= O\left(\sum_{s \neq s_1^*} \max_{k:s \notin S_k^*} \frac{\Delta_{max} L^2 \log T}{\Delta_{s,k}^2}\right).
\end{aligned}
\tag{11}
$$

*for any $\epsilon$ such that $\forall s \neq S_1^*$ and $i \notin S_k^*, \Delta_{s,k} > 2LK\epsilon$, where $U$ is a universal constant.*

**Proof.** For any arm $s \neq s_1^*$, we apply the exploration price $F(i)$ as

$$F(s) = \max_{k:s \notin S_k^*} \frac{6L^2 \log T}{(\Delta_{s,k} - 2LK\epsilon)^2} \tag{12}$$

Therefore, at the $t$-th iteration, there are two events

$$A_t := \left\{\exists i \in \{1, \ldots, \langle W \rangle\} \Rightarrow |\xi_{t,i} - \hat{\mu}_{t,i}|^2 > \frac{3 \log T}{2N_{t,i}}\right\} \tag{13}$$

$$B_t := \left\{\exists i \in \{1, \ldots, \langle W \rangle\} \Rightarrow |\mu_i - \hat{\mu}_{t,i}|^2 > \frac{3 \log T}{2N_{t,i}}\right\}, \tag{14}$$

where $N_{t,s} = \sum_{\tau=1}^{t} \mathbf{1}_{s \in S_\tau}$ is the number of oberservations of arm $s$ and $\hat{\mu}_{t,s} = \frac{1}{N_{t,s}} \sum_{\tau=1}^{t} \sum_{s \in S_\tau} X_{\tau,s}$ is the empirical mean outcome of $s$ at the start of round $t$.

Based on Eq. 13 and 14, the regret $Reg(T)$ can be devided into three terms as

$$Reg(T) \leq \mathbb{E}\left[\sum_{t=1}^{T} \mathbf{1}_{\neg A_t \wedge \neg B_t} \Delta_{S_t}\right] + \mathbb{E}\left[\sum_{t=1}^{T} \mathbf{1}_{A_t} \Delta_{S_t}\right] + \mathbb{E}\left[\sum_{t=1}^{T} \mathbf{1}_{B_t} \Delta_{S_t}\right]. \tag{15}$$

We provide the Lemma C.1, C.2 and C.3 to bound these three terms one by one. Therefore, we can obtain

$$Reg(T) \leq \sum_{s \neq s_1^*} \max_{k:s \notin S_k^*} \frac{6\Delta_s^{\max} L^2 \log T}{(\Delta_{s,k} - 2LK\epsilon)^2} + \left(2K + 4\langle W \rangle + \frac{KU + 8K\epsilon^4}{\epsilon^6}\right) \Delta_{max} \tag{16}$$

### C.2.1 Lemmas

**Lemma C.1.** *Under all assumptions and settings in Theorem 3.4, we have*

$$\mathbb{E}\left[\sum_{t=1}^{T}\mathbf{1}_{\neg A_t \wedge \neg B_t}\Delta_{S_t}\right] \leq \sum_{s\neq s_1^*}\max_{k:s\notin S_k^*}\frac{6\Delta_s^{\max}L^2\log T}{(\Delta_{s,k}-2LK\epsilon)^2} + \left(2K + \frac{KU+8K\epsilon^4}{\epsilon^6}\right)\Delta_{max} \quad (17)$$

**Proof.** To bound this term, we should study the difference between $s_{t,k}$ and $s_k^*$, that is, this term can be bounded by

$$\mathbb{E}\left[\sum_{t=1}^{T}\mathbf{1}_{\neg A_t \wedge \neg B_t}\Delta_{S_t}\right] \leq \sum_{k=1}^{K}\mathbb{E}\left[\sum_{t=1}^{T}\Delta_{S_t}\cdot\mathbf{1}_{\neg A_t \wedge \neg B_t \wedge C_{t,k}}\right] + \sum_{k=1}^{K}\mathbb{E}\left[\sum_{t=1}^{T}\mathbf{1}_{s_k^*=s_{t,k}\wedge||\xi_{t,s_k^*}-\mu_{s_k^*}||_\infty\leq\epsilon}\right]\cdot\Delta_{\max},$$
$$(18)$$

where $C_{t,k}$ denotes as an event which is defined as

$$C_{t,k} := \{S_{k-1}^* = S_{t,k-1} \wedge s_k^* \neq s_{t,k} \wedge ||\xi_{t,S_{k-1}^*}-\mu_{S_{k-1}^*}||_\infty \leq \epsilon\}. \quad (19)$$

We define $\xi_{t,S} = \{\xi_{t,i}|i\in S\}$ and $\mu_S = \{\mu_i|i\in S\}$. According to Lemma C.4, the first term in Eq. 18 can be bounded by:

$$\sum_{k=1}^{K}\mathbb{E}\left[\sum_{t=1}^{T}\Delta_{S_t}\cdot\mathbf{1}_{\neg A_t \wedge \neg B_t \wedge C_{t,k}}\right] \leq \sum_{k=1}^{K}\mathbb{E}\left[\sum_{s\notin S_k^*}\sum_{t=1}^{T}\Delta_{S_t}\cdot\mathbf{1}_{s=s_{t,k}\wedge N_{t,s}\leq F(s)}\right] + \frac{UK}{\epsilon^6}\Delta_{max}$$

$$\leq \mathbb{E}\left[\sum_{s\neq s_1^*}\sum_{t=1}^{T}\sum_{k=1}^{K}\Delta_{S_t}\cdot\mathbf{1}_{s=s_{t,k}\wedge N_{t,s}\leq F(s)}\right] + \frac{UK}{\epsilon^6}\Delta_{max}$$

$$\leq \mathbb{E}\left[\sum_{s\neq s_1^*}\sum_{t=1}^{T}\Delta_{S_t}\cdot\mathbf{1}_{s\in S_t\wedge N_{t,s}\leq F(s)}\right] + \frac{UK}{\epsilon^6}\Delta_{max}$$

$$\leq \sum_{s\neq s_1^*}F(s)\Delta_s^{\max} + \frac{UK}{\epsilon^6}\Delta_{max}$$

$$\leq \sum_{s\neq s_1^*}\max_{k:s\notin S_k^*}\frac{6\Delta_s^{\max}L^2\log T}{(\Delta_{s,k}-2LK\epsilon)^2} + \frac{UK}{\epsilon^6}\Delta_{max},$$
$$(20)$$

where $U$ is a universal constant. Based on Lemma C.5, the second term can be bounded by

$$\sum_{k=1}^{K}\mathbb{E}\left[\sum_{t=1}^{T}\mathbf{1}_{s_k^*=s_{t,k}\wedge||\xi_{t,s_k^*}-\mu_{s_k^*}||_\infty\leq\epsilon}\right]\Delta_{\max} \leq 2K + \frac{8K}{\epsilon^2}\Delta_{\max}. \quad (21)$$

Combining Eq. 20 and Eq. 21, we have

$$\mathbb{E}\left[\sum_{t=1}^{T}\mathbf{1}_{\neg A_t \wedge \neg B_t}\Delta_{S_t}\right] \leq \sum_{s\neq s_1^*}\max_{k:s\notin S_k^*}\frac{6\Delta_s^{\max}L^2\log T}{(\Delta_{s,k}-2LK\epsilon)^2} + \left(2K + \frac{KU+8K\epsilon^4}{\epsilon^6}\right)\Delta_{max}. \quad (22)$$

**Lemma C.2.** *Under all assumptions and setting in Theorem 3.4, we have*

$$\mathbb{E}\left[\sum_{t=1}^{T}\mathbf{1}_{A_t}\Delta_{S_t}\right] \leq 2\Delta_{\max}\langle W\rangle. \quad (23)$$

**Proof.**

$$\mathbb{E}\left[\sum_{t=1}^{T}\mathbf{1}_{A_t}\Delta_{S_t}\right] \leq \mathbb{E}\left[\sum_{t=1}^{T}\mathbf{1}_{A_t}\right]\Delta_{\max}$$

$$\leq \sum_{i=1}^{\langle W\rangle}\mathbb{E}\left[\sum_{t=1}^{T}\mathbf{1}_{|\xi_{t,i}-\hat{\mu}_{t,i}|^2>\frac{3\log T}{2N_{t,i}}}\right]\Delta_{\max}$$

$$\leq \sum_{i=1}^{\langle W\rangle}\sum_{t=1}^{T}\sum_{\tau=1}^{T-1}\mathbb{P}\left(N_{t,i}=\tau \wedge |\xi_{t,i}-\hat{\mu}_{t,i}|^2>\frac{3\log T}{2N_{t,i}}\right)\Delta_{\max}$$

$$= \sum_{i=1}^{\langle W\rangle}\sum_{t=1}^{T}\sum_{\tau=1}^{T-1}\mathbb{P}\left(N_{t,i}=\tau\right)\cdot\mathbb{P}\left(|\xi_{t,i}-\hat{\mu}_{t,i}|^2>\frac{3\log T}{2N_{t,i}}\,\bigg|\,N_{t,i}=\tau\right)\Delta_{\max}. \tag{24}$$

According to Lemma C.6, we have

$$\sum_{i=1}^{\langle W\rangle}\sum_{t=1}^{T}\sum_{\tau=1}^{T-1}\mathbb{P}\left(N_{t,i}=\tau\right)\cdot\mathbb{P}\left(|\xi_{t,i}-\hat{\mu}_{t,i}|^2>\frac{3\log T}{2N_{t,i}}\,\bigg|\,N_{t,i}=\tau\right)\Delta_{\max}$$

$$\leq \sum_{i=1}^{\langle W\rangle}\sum_{t=1}^{T}\sum_{\tau=1}^{T-1}\mathbb{P}\left(N_{t,i}=\tau\right)\cdot 2\exp(-3\log T)\Delta_{\max} \tag{25}$$

$$\leq \sum_{i=1}^{\langle W\rangle}\sum_{t=1}^{T}\frac{2}{T}\Delta_{\max}$$

$$= 2\Delta_{\max}\langle W\rangle.$$

**Lemma C.3.** *With all assumptions and setting in Theorem 3.4, we have*

$$\mathbb{E}\left[\sum_{t=1}^{T}\mathbf{1}_{B_t}\Delta_{S_t}\right] \leq 2\Delta_{\max}\langle W\rangle. \tag{26}$$

**Proof.**

$$\mathbb{E}\left[\sum_{t=1}^{T}\mathbf{1}_{B_t}\Delta_{S_t}\right] \leq \mathbb{E}\left[\sum_{t=1}^{T}\mathbf{1}_{B_t}\right]\Delta_{\max}$$

$$\leq \sum_{i=1}^{\langle W\rangle}\mathbb{E}\left[\sum_{t=1}^{T}\mathbf{1}_{|\mu_i-\hat{\mu}_{t,i}|^2>\frac{3\log T}{2N_{t,i}}}\right]\Delta_{\max}$$

$$\leq \sum_{i=1}^{\langle W\rangle}\sum_{t=1}^{T}\sum_{\tau=1}^{T-1}\mathbb{P}\left(N_{t,i}=\tau \wedge |\mu_i-\hat{\mu}_{t,i}|^2>\frac{3\log T}{2N_{t,i}}\right)\Delta_{\max} \tag{27}$$

$$= \sum_{i=1}^{\langle W\rangle}\sum_{t=1}^{T}\sum_{\tau=1}^{T-1}\mathbb{P}\left(N_{t,i}=\tau\right)\cdot\mathbb{P}\left(|\mu_i-\hat{\mu}_{t,i}|^2>\frac{3\log T}{2N_{t,i}}\,\bigg|\,N_{t,i}=\tau\right)\Delta_{\max}.$$

According to Lemma C.7, we have

$$\sum_{i=1}^{\langle W\rangle}\sum_{t=1}^{T}\sum_{\tau=1}^{T-1}\mathbb{P}\left(N_{t,i}=\tau\right)\cdot\mathbb{P}\left(|\mu_i-\hat{\mu}_{t,i}|^2>\frac{3\log T}{2N_{t,i}}\,\bigg|\,N_{t,i}=\tau\right)\Delta_{\max}$$

$$\leq \sum_{i=1}^{\langle W\rangle}\sum_{t=1}^{T}\sum_{\tau=1}^{T-1}\mathbb{P}\left(N_{t,i}=\tau\right)\cdot 2\exp(-3\log T)\Delta_{\max} \tag{28}$$

$$\leq \sum_{i=1}^{\langle W\rangle}\sum_{t=1}^{T}\frac{2}{T}\Delta_{\max}$$

$$= 2\Delta_{\max}\langle W\rangle.$$

**Lemma C.4.** *Under all assumptions and settings in Theorem 3.4, with $C_{t,k}$ defined in Eq. 19, we have*

$$\mathbb{E}\left[\sum_{t=1}^{T}\Delta_{S_t}\cdot\mathbf{1}_{\neg A_t\wedge\neg B_t\wedge C_{t,k}}\right]\leq\mathbb{E}\left[\sum_{s\notin S_k^*}\sum_{t=1}^{T}\Delta_{S_t}\cdot\mathbf{1}_{s=s_{t,k}\wedge N_{t,s}\leq F(s)}\right]+\frac{U\Delta_{max}}{\epsilon^6}. \qquad (29)$$

**Proof.** This Lemma is a special case in [26] (Lemma 1), setting the size of the unit as 1 and $U = CC'$.

**Lemma C.5.** *With all assumptions and setting in Theorem 3.4, we have*

$$\mathbb{E}\left[\sum_{t=1}^{T}\mathbf{1}_{s_k^*=s_{t,k}\wedge||\xi_{t,s_k^*}-\mu_{s_k^*}||_\infty\leq\epsilon}\right]\Delta_{\max}\leq 2+\frac{8}{\epsilon^2}\Delta_{\max} \qquad (30)$$

**Proof.** This Lemma is a special case in [26] (Lemma 3), setting the size of each unit as 1.

**Lemma C.6.** *With all assumptions and setting in Theorem 3.4, for any arm $i\in\{1,\cdots,\langle W\rangle\}$ and round $t$, we have*

$$\mathbb{P}\left(|\xi_{t,i}-\hat{\mu}_{t,i}|^2>\epsilon\big|\alpha_{t,i},\beta_{t,i}\right)\leq 2\exp(-2\epsilon N_{t,i}), \qquad (31)$$

*where $\alpha_{t,i}$ and $\beta_{t,i}$ are the values of $\alpha_t$ and $\beta_t$ in the TSAdj before the start of round $t$*

**Proof.** This Lemma has been proved in [59] (Lemma 3).

**Lemma C.7.** *With all assumptions and setting in Theorem 3.4, let $X_1,\cdots,X_K$ be indentical independent random variable such that $X_i\in[0,1]$ and $\mathbb{E}[X_i]=\mu$ for any $i\in\{1,\ldots,K\}$. Then for any $\epsilon\geq 0$, we have*

$$\mathbb{P}\left(\left|\sum_{i=1}^{K}X_i-\mu\right|>K\epsilon\right)\leq 2\exp(-2K\epsilon^2), \qquad (32)$$

**Proof.** This Lemma can be proved by Chernorff-Hoeffding Bound in [44](Theorem 1.1).

# D  Broader Impact

Beyond its technical contributions, FedRTS holds significant practical potential for democratizing efficient federated learning. By substantially reducing communication and computation costs via dynamic sparse training, our approach could broaden the adoption of FL in resource-constrained environments, such as the Internet of Things [31, 4, 2, 3, 58], edge perception [61, 62, 60], and privacy-sensitive medical applications [16]. However, like all pruning-based methods, FedRTS carries a risk of inadvertently amplifying biases if sparsity patterns disproportionately discard weights that encode features of minority classes. To mitigate this, we recommend integrating our method with fairness-aware regularization and continual learning techniques [45, 63] in sensitive domains. Future work will explicitly investigate the societal implications of deploying FedRTS at scale, with a focus on applications where model sparsity may impact predictive reliability.

# E  Additional Experiments Results

To further validate our findings, we perform additional experiments to investigate the impact of various hyperparameters and the convergence behavior of our proposed methods.

## E.1  The Impact of Hyperparameters

### E.1.1  The Impact of trade-off ratio $\gamma$

The trade-off ratio $\gamma$ modulates the weight of individual models' information. Traditional methods only incorporate the information from the aggregated model, which is insufficient. To verify the impact of individual models' information, we conduct experiments on the trade-off ratio $\gamma$. As illustrated in Fig. 7, the test accuracy reaches its peak when the $\gamma$ value is between 0.3 and 0.5. This indicates that the information from both individual models and the aggregated model is significant in model topology adjustment.

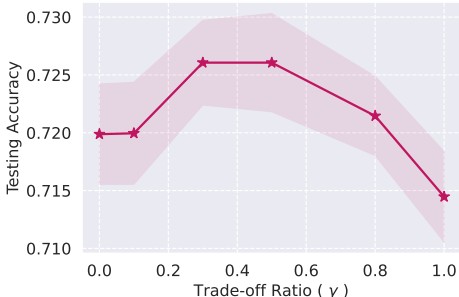

Figure 7: Impact of trade-off ratio $\gamma$

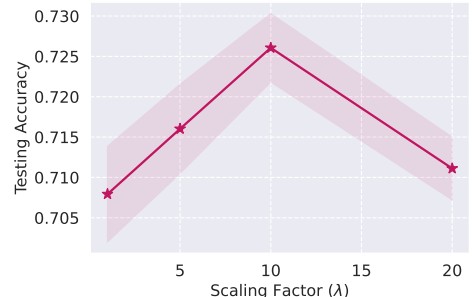

Figure 8: Impact of the scaling factor $\lambda$

| $\Delta T$ | 3 | 5 | 10 | 20 |
|---|---|---|---|---|
| FedDST | 0.664 | 0.683 | **0.723** | **0.716** |
| FedMef | **0.745** | **0.737** | 0.713 | 0.705 |
| **FedRTS** | **0.735** | **0.722** | **0.734** | **0.731** |

Table 2: Performance under Different Adjustment Interval ($\Delta T$)

|  | CIFAR-10 | | CINIC-10 | |
|---|---|---|---|---|
|  | FLOPs | Comm. Cost | FLOPs | Comm. Cost |
| FedDST | 3.4E16 | 221.9 GB | 8.2E16 | 290.7 GB |
| FedMef | 3.5E16 | 227.6 GB | 6.1E16 | 222.3 GB |
| FedRTS | 3.1E16 | 204.2 GB | 2.9E16 | 105.6 GB |

Table 3: The overall training FLOPs and communicational cost (Comm. Cost) for FedRTS and other frameworks to reach the accuracy of dense FedAVG.

### E.1.2 The Impact of the scaling factor $\lambda$

The scaling factor $\lambda$ regulates the impact of the current reward $X(t)$ at step $t$. A $\lambda$ value that is too small will amplify the uncertainty in Thompson Sampling, whereas a $\lambda$ value that is too large will disregard the uncertainty arising from insufficient samples. Thus, selecting an appropriate value for $\lambda$ is crucial for enhancing FedRTS's performance. In a separate set of experiments, the influence of the reward scaling $\lambda$ on the results is investigated. Figure 8 shows that the test accuracy attains its maximum at approximately 0.73 when $\lambda$ is 10.

### E.1.3 The Impact of the number of core links $\kappa$

The number of core links $\kappa$ critically controls the adjustment intensity (the number of weights pruned and reactivated) in each outer loop, creating an essential trade-off: large $\kappa$ values lead to sluggish topology evolution, while small $\kappa$ risks substantial information loss. We formalize this through our decay schedule: $\kappa^l = K^l - \frac{\alpha_{adj}}{2}(1 + \cos(\frac{t\pi}{T_{end}}))K^l$, where $K^l$ is the number of active weights at $l$-th layer. Our empirical investigation on CIFAR-10 in Fig. 13 demonstrates FedRTS's robustness across varying $\alpha_{adj}$, consistently outperforming baseline methods.

### E.1.4 The Impact of Adjustment Interval $\Delta T$

In federated dynamic sparse training, the adjustment interval ($\Delta T$) between two outer loops for adjustment critically influences model performance through a fundamental trade-off. Excessively long intervals may delay the discovery of optimal sparse patterns, while overly frequent adjustments can prevent adequate model adaptation between updates and require higher resource consumption.

Our experiments with FedRTS on CIFAR-10 explored this trade-off by evaluation with various adjustment intervals ($\Delta T \in \{3, 5, 10, 20\}$). As depicted in Table 2, FedRTS consistently outperformed baseline methods across most cases. Notably, FedRTS's performance improved with shorter intervals, demonstrating FedRTS's unique ability to quickly recover from topology changes, which is a crucial advantage in dynamic sparse training scenarios.

## E.2 Efficiency Analysis

### E.2.1 Training Cost

We evaluate FedRTS's efficiency by analyzing training costs on CIFAR-10 and CINIC-10. To compare expenses, we measure total training FLOPs and communication costs required to match the

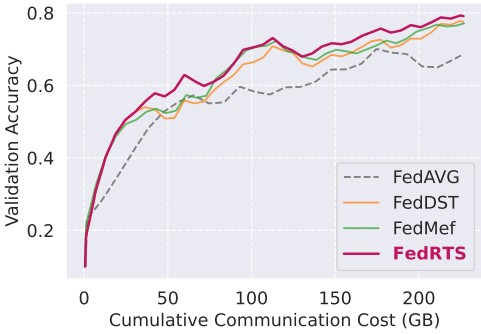

Figure 9: Validation accuracy of FedRTS and SOTA frameworks on CIFAR-10 dataset for cumulative communicational cost.

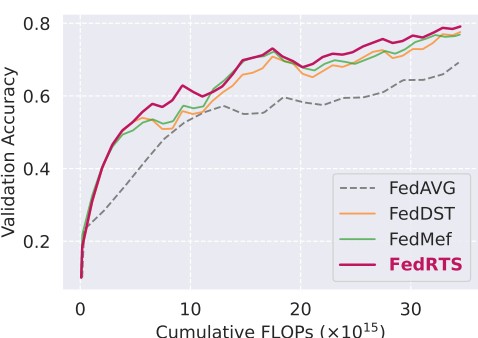

Figure 10: Validation accuracy of FedRTS and SOTA frameworks on CIFAR-10 dataset for cumulative FLOPs during training.

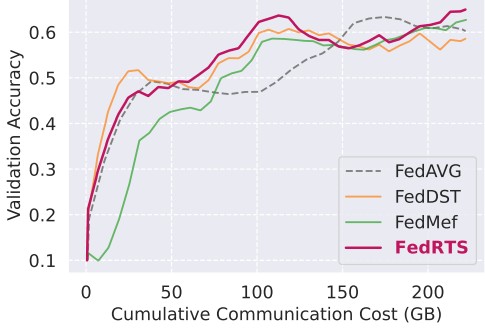

Figure 11: Validation accuracy of FedRTS and SOTA frameworks on CINIC-10 dataset for cumulative communicational cost during training.

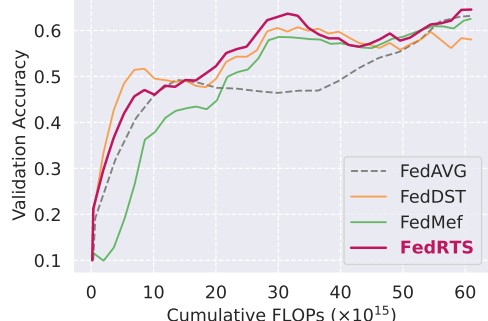

Figure 12: Validation accuracy of FedRTS and SOTA frameworks on CINIC-10 dataset for cumulative FLOPs during training.

testing accuracy of dense FedAVG. The accuracy is averaged over 10 rounds to mitigate variance from random factors, ensuring a more stable evaluation.

As shown in Tab. 3, FedRTS achieves the testing accuracy of dense FedAVG with significantly lower costs. On CINIC-10, it achieves 48% training FLOPs and the communication cost compared to the best baseline FedMef, which highlights the efficiency of FedRTS.

### E.2.2 Convergence Behavior

Convergence is crucial for effective learning, precise predictions, and optimized resource use. To evaluate this, we track the validation accuracy of FedRTS and SOTA frameworks on CIFAR-10 and CINIC-10 against cumulative communication and computational costs, as shown in Fig. 9, Fig. 10, Fig. 11, and Fig. 12. Accuracy is averaged over 10 rounds to reduce variance and provide a stable representation. Our experimental results demonstrate distinct convergence patterns across datasets. On CIFAR-10, FedRTS exhibits superior convergence characteristics throughout the entire training process. The CINIC-10 dataset reveals a more complex trajectory: while FedDST initially demonstrates higher performance during early training phases, FedRTS consistently surpasses competing methods in later stages when considering both communication efficiency and computational costs. These empirical results provide strong evidence for the effectiveness of the TSAdj mechanism in FedRTS, particularly in optimizing the trade-off between model performance and resource consumption during federated training.

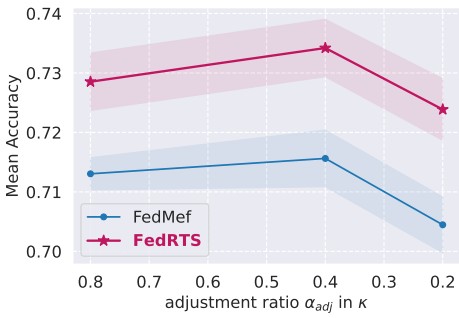
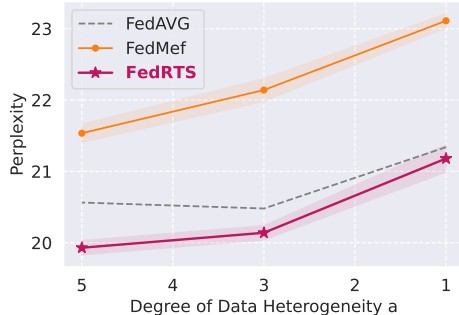

Figure 13: Accuracy on CIFAR-10 with ResNet under various degrees of adjustment ratio $\alpha_{adj}$.

Figure 14: Perplexity on TinyStories under various degrees of data heterogeneity. The lower $a$ represents a higher non-IID degree.

## E.3 Broader Experiments

### E.3.1 FedRTS vs. FedCUCB

The Combinatorial Upper Confidence Bound (CUCB) is another optimization method for the Combinatorial Multi-Armed Bandit (CMAB) problem. Unlike Combinatorial Thompson Sampling (CTS), which relies on probabilistic sampling, CUCB employs deterministic optimistic reward estimates to balance exploration and exploitation. However, its inherent optimism can lead to persistently selecting suboptimal arm combinations, particularly in dynamic pruning scenarios where adaptability is crucial.

CUCB maintains upper confidence bounds (UCBs) rather than probability distributions for each arm $i$, which formulated as

$$\bar{\mu}_{t,i} = \hat{\mu}_{t,i} + \sqrt{\frac{3\log t}{2\Psi_i}}, \tag{33}$$

where $\Psi_i$ denotes the number of times arm $i$ has been selected, $\hat{\mu}_{t,i}$ denotes the empirical mean reward (initialized as $\hat{\mu}_{0,i} = 0$). The selected arms for action $S_t$ are determined by

$$S_t = \{i \in \text{Top}(\bar{\mu}_t, K)\}$$

For the selected arms $i \in S_t$ with outcomes $X_t$, the empirical mean is updated as:

$$\Psi_i \leftarrow \begin{cases} \Psi_i, & i \notin S_t \\ \Psi_i + 1, & i \in S_t \end{cases}, \quad \hat{\mu}_{t+1,i} = \begin{cases} \hat{\mu}_{t,i}, & i \notin S_t \\ \frac{\hat{\mu}_{t,i}(\Psi_i-1)+X_{t,i}}{\Psi_i}, & i \in S_t \end{cases}, \tag{34}$$

The next round's action selection then uses the updated UCB $\bar{\mu}_{t+1,i}$ computed by Eq. 33.

A notable drawback of FedCUCB is its handling of under-explored ("bad") arms. If an arm $i$ is rarely selected (low $\Psi_i$), its confidence bound $\bar{\mu}_{t,i}$ grows indefinitely as $t \to \infty$, eventually forcing its selection. This behavior disrupts convergence, as the algorithm repeatedly revisits suboptimal arms instead of refining the topology.

To evaluate this limitation, we propose FedCUCB, a federated pruning framework based on CUCB. In contrast to FedRTS, FedCUCB maintains UCB $\bar{\mu}_t$ and define the outcomes $X_t$ as same in Eq. 3. The results in Fig. 15 demonstrates that FedCUCB performs slightly worse than FedRTS, but still outperform other baselines.

### E.3.2 Pruning vs. Weights Quantization

Parallel to pruning-based approaches, quantization techniques [50, 23, 15, 17, 20] offer an alternative for reducing communication overhead in FL. While both strategies aim to alleviate transmission costs,

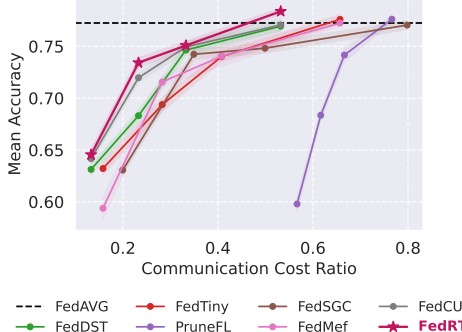

| Method | Comm. Cost | FLOPs | Memory | Acc. |
|--------|-----------|-------|--------|------|
| FedAVG | 89.52 MB | 17.8E+12 | 134.29MB | 77.24 |
| FedPAQ | 22.48 MB | 17.8E+12 | 134.29 MB | **73.34** |
| FedDST | **20.85 MB** | **3.8E+12** | **31.28 MB** | 68.31 |
| FedRTS | **20.85 MB** | **3.8E+12** | **31.28 MB** | **73.41** |

Table 4: Performance and training cost comparison between federated pruning (FedDST, FedRTS) and federated quantization (FedPAQ) frameworks on CIFAR-10 using ResNet18. FedPAQ apply 8-bit quantization, while FedDST and FedRTS employ a target density of $d' = 0.2$.

Figure 15: Accuracy on CIFAR-10 with ResNet18 for FedRTS, FedCUCB and other federated dynamic pruning methods.

weights quantization operates after dense local training, whereas pruning enables sparse training by design. This fundamental difference grants pruning distinct advantages in resource-constrained settings: it reduces computation through selective updates, lowers memory usage by maintaining dynamic sparsity, and ensures consistent communication efficiency across rounds.

Empirical evidence supports these trade-offs. As shown in Table 4, our FedRTS outperforms 8-bit FedPAQ [50], a weights quantization baseline, in computational and memory efficiency, while retaining competitive model accuracy. This makes pruning a more holistic solution for FL deployments on edge devices.

# F   Additional Setup Details

## F.1   Datasets

In the CV tasks, we apply our approach using ResNet18 on CIFAR-10, CINIC-10, TinyImageNet, and SVHN datasets.

- CIFAR-10: It contains 50,000 training images and 10,000 testing images. Each image in CIFAR-10 is a 3×32×32 RGB image, implying 10 classes of objects.
- CINIC-10: It contains three equal subsets - train, validation, and test–each comprising 90,000 images. CINIC-10 is an extended version of CIFAR-10 that includes additional images from ImageNet.
- TinyImageNet: It contains 200 classes with 500 training images, 50 validation images, and 50 test images per class, each resized to 64×64 pixels.
- SVHN: There are 73,257 digit images in the training set and 26,032 images in the testing set. The digit images are obtained from house numbers in Google Street View images.

In the NLP tasks, we choose GPT-2-32M as the backbone model with the TinyStories dataset. TinyStories contains synthetically generated short stories that only use a limited vocabulary. We truncate each story to 256 tokens, with 2,120,000 stories used for training and 22,000 for testing.

## F.2   Baselines

The selected federated pruning baselines differ in their approaches to sparse model initialization and subsequent model topology adjustment:

- PruneFL: Initializes the sparse model by training on a powerful device and applies adaptive pruning by importance measure. In our implementation, we perform random pruning instead of pruning with powerful devices due to resource constraints. Moreover, we set the time $t_j$ of each parameter $j$ with 1.

- FedDST: Applies dynamic sparse training on clients and performs greedy aggregation to produce a new global topology.
- FedTiny: Initializes the model by adaptive batch normalization selection and adjusts the model mask based on the aggregated top-K gradients of parameters.
- FedSGC: Adjusts model topology on devices guided by the gradient congruity, which is similar to FedDST but requires extra communication cost.
- FedMef: Utilizes budget-aware extrusion to transfer essential information of pruned parameters to other parameters and reduces activation memory by scaled activation pruning.

## F.3 Extra Information

The experiments are simulated via multi-process on an Nvidia RTX 5880 GPU, an Intel Core i9 CPU with 48 GB of memory.

## F.4 Communication and Computational Cost

In our experiments, we analyzed the proposed FedRTS against other methods by examining the computational FLOPs and communication costs. We began by introducing sparse compression strategies and then detailed how we calculated these metrics.

### F.4.1 Compression Strategy

When storing a matrix, two key components are values and positions. Compression techniques focus on reducing the storage required for the positions of non-zero values within the matrix. Consider a scenario where we aim to store the positions of m non-zero values with a b-bit width in a sparse matrix M, which comprises n elements and has a shape of nr × nc. The density of the matrix, denoted by d = m/n, determines the compression scheme applied to represent M. The storage involves using o bits to represent the positions of the m non-zero values, resulting in an overall storage size of s.

- For density ($d \in [0.9, 1]$), a dense scheme is utilized, leading to $s = nb$.
- For density $d \in [0.3, 0.9)$, the bitmap (BM) method is employed, storing a map with $n$ bits, where $o = n$ and $s = o + mb$.
- For density $d \in [0.1, 0.3)$, the coordinate offset (COO) scheme is applied, storing elements with their absolute offsets and requiring $o = m\lceil log2n \rceil$ extra bits for position storage. Thus, the overall storage becomes $s = o + mb$.
- For density $d \in [0., 0.1)$, the compressed sparse row (CSR) and compressed sparse column (CSC) methods are used based on size considerations. These methods use column and row indexes to store element positions, with CSR needing $o = m\lceil log_2 n_c \rceil + n_r \lceil log_2 m \rceil$ bits. The overall storage size is $s = o + mb$.

Reshaping is performed on tensors before compression, enabling the determination of the memory required for training the network's parameters.

### F.4.2 Communication Cost

In terms of communication costs, FedRTS shares a communication cost strategy similar to that of FedTiny. In contrast to other baseline methods like FedDST, where clients need to upload TopK gradients to the server every $\Delta R$ rounds to facilitate parameter expansion, the quantity of TopK gradients, denoted as $\xi_t$, aligns with the number of marked parameters $\theta$ low. Here, $\xi_t = \zeta_t(1-s_m)n_\theta$, with $\zeta_t = 0.2(1 + cos \frac{t\pi}{R_{stop}E})$ representing the adjustment rate for the t-th iteration. This results in minimal upload overhead. Moreover, there is no communication overhead for the model mask m during download since sparse storage formats like bitmap and coordinate offset contain the same element position information. Auxiliary data such as the learning rate schedule are excluded.

We define the storage for dense and sparse parameters as $O_d$ and $O_s$, respectively. Notably, in the inner loop, all federated pruning methods except FedAVG share the same communication cost, which amounts to $2O_s$. However, in the outer loop, different federated learning frameworks exhibit varying communication costs. The communication costs of different federated learning frameworks in the outer loop are detailed below:

- FedAVG: The data exchange amounts to $2O_d$, encompassing the uploading and downloading of dense parameters.

- FedDST: In this scenario, the model mask does not necessitate additional space for storage as the compressed sparse parameters already embody the mask information. Hence, the data exchange per round stands at $2O_s$, covering the upload and download of sparse parameters.

- PruneFL: PruneFL requires the clients to upload the full-size gradients squared for adjustment, therefore, the data exchange in the outer loop is $O_d + O_s$.

- FedSGC: FedSGC requires the server to send the gradient congruity to the server for each round, which has the same size as sparse weight. Therefore, the data exchange in the outer loop is $3O_s$.

- FedTiny and FedMef: In comparison to FedDST, FedTiny and FedMef entail the upload of TopK gradients every $\Delta R$ rounds. Therefore, the maximum data exchange per round sums up to $2O_s + O_\xi$, where $O_\xi$ signifies the storage required for the TopK gradients.

- FedRTS: When compared to FedTiny and FedMef, FedRTS only requires uploading the index of the TopK gradients every $\Delta R$ rounds, while FedTiny and FedMef necessitate uploading both the index and the values of TopK gradients at the same intervals. Consequently, the maximum data exchange per round amounts to $2O_s + 0.5O_\xi$, where $O_\xi$ represents the storage needed for the TopK gradients.

### F.4.3  Computational FLOPs

Training FLOPs encompass both forward-pass FLOPs and backward-pass FLOPs, with operations tallied on a per-layer basis. During the forward pass, layer activations are sequentially computed using prior activations and layer parameters. In the backward pass, each layer computes activation gradients and parameter gradients, with twice as many FLOPs expended in the backward pass as in the forward pass. FLOPs related to batch normalization and loss calculation are excluded.

In a detailed breakdown, assuming inference FLOPs for dense and static sparse models are denoted as $F_d$ and $F_s$ respectively, and the local iteration count is $E$, the maximum training FLOPs for each framework are as follows:

- FedAVG: Requires training a dense model, resulting in training FLOPs per round amounting to $3F_d E$.

- PruneFL: it requires calculating the dense gradients for each backward, therefore the training FLOPs is $(2F_d + F_s)E$.

- FedMef: Introduces a minor calculation overhead for BaE and SAP. Therefore, the maximum training FLOPs are estimated as $3(F_s + F_o)(E - 1) + (F_s + F_o) + 2F_d$, where Fo represents the computational overhead of BaE and SAP. Fo is approximated as $F_o = 4(1 - s_m)n_\theta + n_a log_{n_a}$, with $4(1 - s_m)n_\theta$ representing the FLOPs for regularization and WSConv, and $n_a log n_a$ indicating the FLOPs for activation pruning.

- FedRTS, FedTiny, FedSGC, and FedDST: Implement RigL-based methods to adjust model architectures, necessitating clients to compute dense gradients in the final iteration. The maximum training FLOPs sum up to $3F_s(E - 1) + F_s + 2F_d$.

