# OpenReview forum: "FedRTS: Federated Robust Pruning via Combinatorial Thompson Sampling"
_NeurIPS.cc/2025/Conference — NeurIPS 2025 poster_

### Official Review · Reviewer_1eQR · 2025-06-22

**Clarity:** 2
**Significance:** 3
**Originality:** 3
**Rating:** 3
**Confidence:** 3

**Summary:**

This paper addresses challenges such as instability introduced by traditional pruning methods in federated learning and further proposes a novel federated pruning approach based on Thompson Sampling. The authors conduct comprehensive theoretical analysis and empirical experiments to demonstrate the superiority of their method.

**Questions:**

1. When defining $f_n()$ in line 66, the authors could first define the loss function, then define the empirical risk or expected risk as the objective.
2. Please provide the precise mathematical definition of the density $d$ mentioned in line 66.
3. What does the symbol $\mathbf{1}$ in line 84 represent?
4. In line 87, what exactly is $X_t$ in the reward? Given that the authors mention it has a "mean"—please provide a precise definition of this random variable and clarify the source of its randomness.
5. In line 125, does maintaining posterior probabilities for each link in TSAdj incur excessive computational overhead?
6. Although the authors provide a regret analysis for the proposed pruning method, can they also provide a convergence analysis of the global model trained using this pruning method?
7. The theoretical contributions of the paper are relatively weak. Please elaborate on the technical challenges involved in analyzing the regret of the Thompson Sampling algorithm in the federated learning pruning scenario considered in this paper.
8. It is recommended that the authors consider more heterogeneous data splitting settings (e.g., concept shift or covariate shift) to evaluate the robustness of their proposed FedRTS method.

**Ethical Concerns:**

["NO or VERY MINOR ethics concerns only"]

**Final Justification:**

My final decision is to maintain my initial score. While the authors have provided a rebuttal outlining the contributions of the paper, I still argue that the work primarily applies Thompson sampling to federated pruning without formulating or addressing any fundamental problems, nor introducing new techniques.

**Limitations:**

yes

**Paper Formatting Concerns:**

There are no major formatting issues in this paper.

**Quality:**

3

**Strengths And Weaknesses:**

**Strengths:** This paper conducts very comprehensive experiments, validating the proposed methods across various model sizes and tasks in both the CV and NLP domains.


**Weaknesses:** The authors do not appear to provide an in-depth analysis of the overhead of the sampling procedure in FedRTS, which may entail significant costs. Some notations and definitions are unclear (see specific questions for details). Additionally, the theoretical contributions are relatively weak.

---

> ### Author Rebuttal · Authors · 2025-07-31
>
> We appreciate Reviewer 1eQR's insightful feedback, especially recognizing that our paper conducts very comprehensive experiments. We will address all concerns below.
>
> > **Q1** When defining $f_n$ in line 66, the authors could first define the loss function, then define the empirical risk or expected risk as the objective.
>
> The loss function $f_n$ depends on the task of the $n$-th device, e.g., cross-entropy loss for the image classification task.
>
> > **Q2** Please provide the precise mathematical definition of the density mentioned in line 66.
>
> The density $d$ is defined as the ratio of non-zero elements to all elements, i.e., $d = \frac{sum(m)}{\langle m \rangle}$, where $m$ denotes the binary mask, $sum(m)$ is the number of non-zero entries, and $\langle m \rangle$ is the total number of elements of $m$.
>
> > **Q3** What does the symbol **1** in line 84 represent?
>
> **1** denotes an indicator function, where $\mathbf{1_{true}}= 1$, $\mathbf{1_{false}}= 0$. We apologize for the mistake in line 84: it should be $m_{t,i} = \mathbf{1}_{i \in S_t}$.
>
> > **Q4** What exactly is $X_t$ in the reward? Please provide a precise definition of this random variable and clarify the source of its randomness.
>
> $X_t$ denotes the outcome (empirical importance) observed with selected arms $S_t$. For the active arm $i \in S_t$, the $[X_t]_i$ is the probability of being among the top weight magnitudes after local training. For the inactive arm $i \notin S_t$, the $[X_t]_i$ is the probability of being among the top gradient magnitudes after local training. These probabilities are computed through a voting mechanism as in Eqs.3, 4, and 5. The randomness of $X_t$ comes from the inherent stochasticity in the training process and random client sampling.
>
> > **Q5** Does maintaining posterior probabilities for each link in TSAdj incur excessive computational overhead?
>
> In TSAdj's design, maintaining posterior probabilities **does not incur excessive overhead**. On the server, weights aggregation iterates exactly $\langle C_t \rangle$ times, requiring $\mathcal{O}(\langle C_t \rangle \langle W \rangle)$ for weight averaging, $\mathcal{O}(\langle C_t \rangle \langle W \rangle)$ for outcome computation (expected time with Quick Sort), $\mathcal{O}(\langle W \rangle)$ for updating beta distributions and sampling probabilities, $\mathcal{O}(\langle W \rangle) $ for arms selection.
>
> The overall complexity simplifies to $\mathcal{O}(\langle C_t \rangle \langle W \rangle)$. Compared to baselines' complexity (e.g., FedMef/FedTiny/PruneFL's $\mathcal{O}(\langle C_t \rangle \langle W \rangle)$), our proposed TSAdj achieves the same computational complexity, demonstrating that TSAdj introduces no significant additional computational burden on the server.
>
> Most importantly, TSAdj’s overhead is **server-side**, and it has **NO extra computational overhead** compared to baselines on devices. In the cross-device FL, the server is usually assumed to have substantial resources.
>
> > **Q6** Although the authors provide a regret analysis for the proposed pruning method, can they also provide a convergence analysis of the global model trained using this pruning method?
>
> We provide the following non-convex convergence analysis for our FedRTS. We need to add the following common assumptions,
>
> _Assumption 3.4 (Bounded variance of local gradients) There exist constants $\beta$ and $\zeta$ such that_
>
> $\frac{1}{N} \sum_{n=1}^N ||\nabla f_n(W) - \nabla f(W)||^2 \le \beta^2 ||\nabla f(W)||^2 +  \zeta^2$
>
> _Assumption 3.5 (Bounded gradients) The gradients are bounded by a constant $\psi$, i.e., $\mathbb{E}[||\nabla f(W)||^2] \le \psi^2$._
>
> _Assumption 3.6 (Bounded weights) the weight magnitudes are bounded by a constant $\varphi$, i.e.,_
>
> $\mathbb{E}[||W||_\infty] \le \varphi$
>
> Assumptions 3.4 and 3.5 follow prior works [1,2], and Assumption 3.6  is justified by the initialization scheme and the regularization term. Based on these assumptions, we establish the convergence upper bound for FedRTS.
>
> __Theorem 2 (Convergence of FedRTS)__. Under Assumptions 3.1 (L-continuity), 3.4, 3.5 and 3.6, for FedRTS with TSAdj with full client participation and arithmetic aggregation, $p_n = 1/N$, with learning rate $\eta \le \frac{1}{2LE\sqrt{3\beta^2+3}}$, adjustment intensity $ \kappa_t = K - \frac{\alpha_{adj}K}{\sqrt{t}}$, the sparse model $W_t \odot m_t$ satisfies the following upper bounds:
>
> $$\min_{1 \le t \le T}  \mathbb{E}\left[||\nabla f (W_t \odot m_t)||^2\right]   \le  \frac{\Omega_1\}{\Delta T\eta \sqrt{T}}+ \frac{\Omega_2}{\eta T} + \Omega_3\eta^2 $$
>
> where $\Omega_1 =\frac{12\alpha_{adj} K(2\psi\varphi + \varphi^2L)}{E}$, $\Omega_2 = \frac{12\Delta f_0 }{E}$, $\Omega_3 = 12L^2E^2\zeta^2$ , $\Delta f_0$ denotes the gap between initial model and optimal, i.e., $\Delta f_0 = f(W_0 \odot  m_0) - f*$, $f*$ is the optimal value. specifically, when $\eta = \frac{1}{2T^{1/6}LE\sqrt{3\beta^2+3}}$, consequently for a given $\varepsilon > 0$ we have that
> $$ T = O(\frac{1}{\varepsilon^{3}\Delta T^3} + \frac{1}{\varepsilon^{\frac{6}{5}}} + \frac{1}{\varepsilon^{3}}) = O(\frac{1}{\varepsilon^3}). $$
>
> __proof.__
>
> Due to limited space, we provide a sketch of proof for the Theorem, and we can provide the proof to Lemma 1 and Lemma 2 in further discussion if the reviewer is interested. With the key Lemma 1 (Bound of topology adjustment) and Lemma 2 (Upper bound of local training), the bound of loss descent in one loop is
>
> $$ \mathbb{E} \left[ f(W_{t+1} \odot m_{t+1}) - f(W_{t} \odot m_{t}) \right]  \le - \frac{1}{12} \eta E \mathbb{E}[||\nabla f(W_{t} \odot m_{t})||^2] + \eta^3 L^2 E^3\zeta^2 + (\psi\varphi + \frac{L\varphi^2}{2})(K-\kappa_t) $$
>
> By rearranging the terms, subtracting $f*$, and denoting the $r_t = \mathbb{E}[f(W_{t} \odot m_t) - F^*]>0$, $\Omega_3 = 12L^2E^2\zeta^2$ , the Inequality converts to
>
> $$\mathbb{E}||\nabla f(w_{t} \odot m_t)||^2 \le \frac{12}{E\eta}(r_{t} - r_{t+1})  + \frac{6(2\psi\varphi + L\varphi^2)}{E\eta}(K - \kappa_t) + \Omega_3\eta^2.$$
>
> After that, given the fact $\min_{1\le t \le T}\mathbb{E}[||\nabla f(W_{t}\odot m_t)||^2] \le \frac{1}{T}\sum_{t=1}^{T}\mathbb{E}[||\nabla f(W_{t}\odot m_t)||^2]$, by summing $t$ from $1$ to $T$, we can obtain
>
> $$ \min_{1 \le t \le T} \mathbb{E}||\nabla f (W_t \odot m_t)||^2  \le
>  \frac{6(2\psi\varphi + L\varphi^2)\sum_{t=1}^{T}(K - \kappa_t)}{E\eta T} + \frac{12r_0}{E\eta T} + \Omega_3\eta^2.$$
>
> Given the definition of $\kappa  = K - \frac{\alpha_{adj}}{2}(1+\cos{(\frac{t\pi}{T_{end}})})K$ is related to $t$ and the adjustment performs every $\Delta T$, we use the fact
>
> $$\sum_{t=1}^{T}(K - \kappa_t) \le \frac{2\sqrt{T}\alpha_{adj}K}{\Delta T}$$
>
> Given $r_0 = \Delta f_0 = f(w_0) - f*$, we can obtain
>
> $$\min_{1 \le t \le T}  \mathbb{E}\left[||\nabla f (W_t \odot m_t)||^2\right]   \le  \frac{\Omega_1\}{\Delta T\eta \sqrt{T}}+ \frac{\Omega_2}{\eta T} + \Omega_3\eta^2 $$
>
> where $\Omega_1 =\frac{12\alpha_{adj} K(2\psi\varphi + \varphi^2L)}{E}$, $\Omega_2 = \frac{12\Delta f_0 }{E}$.
>
> **Lemma 1.** (Upper bound of topology adjustment) Under Assumptions 3.1 (L-continuity), 3.5, and 3.6, in the $t$-th loop, the information loss in the topology adjustment is bounded by
>    $$\mathbb{E}\left[ f(W_{t+1} \odot m_{t+1}) - f(W_{t+1} \odot m_{t}) \right] \le \psi\varphi \sqrt{K - \kappa_t}+ \frac{(K -\kappa_t) L\varphi^2}{2},$$
>
> **Lemma 2.** (Upper bound of local training) Under Assumptions 3.1 (L-continuity) and 3.5, if $\eta \le \frac{1}{2LE\sqrt{3\beta^2+3}}$, $p_n = 1/N$, then
>     $$\mathbb{E} [f(W_{t+1} \odot m_{t}) - f(W_t \odot m_{t})] \le  - \frac{1}{12} \eta E \mathbb{E} \left[||\nabla f(W_{t} \odot m_{t})||^2 \right] + \eta^3 L^2 E^3\zeta^2$$
>
> __Remark.__ The convergence theorem provides valuable insight into how hyperparameters affect convergence. Specifically, a lower adjustment ratio $\alpha_{adj}$, less frequent adjustments (i.e., a larger $\Delta T$) may facilitate faster convergence.
>
> [1] Li, Y., & Lyu, X.. Convergence analysis of sequential federated learning on heterogeneous data. NeuIPS, 2023.
>
> [2]  Li, Xiang, et al. "On the convergence of fedavg on non-iid data." ICLR 2020.
>
> > **Q7** Please elaborate on the technical challenges involved in analyzing the regret of the Thompson Sampling algorithm in the federated learning pruning scenario considered in this paper.
>
> The main difficulty in this paper lies in the __unobservable reward function__. Since the server cannot directly access client data, the true impact of pruning decisions (i.e., the reward) cannot be observed, making it impossible to identify the global optimal arms at each step. Referring to the previous TS work [3], we use the empirical mean reward μ as a proxy for the unknown reward function and treat the greedy optimal arms as the target. Our work provides the __first__ theoretical baseline on federated pruning with TS, and we acknowledge that developing more sophisticated regret analyses for federated settings—particularly under challenges such as non-stationarity—remains an important avenue for future work.
>
> [3] Wang, Siwei, and Wei Chen. "Thompson sampling for combinatorial semi-bandits." ICML, 2018.
>
> > **Q8**  It is recommended that the authors consider more heterogeneous data splitting settings (e.g., concept shift or covariate shift) to evaluate the robustness of their proposed FedRTS method.
>
> We have evaluated FedRTS and FedMef under feature distribution skew (covariate shift). While maintaining quantity and label balance across clients, we injected different types of Gaussian noise into the input features. Each client randomly selected one of two or three types of Gaussian noise, each with distinct means and variances. The results on CIFAR-10 (0.3 density) are shown below:
>
> ||IID|2-Type Noise|3-Type Noise |
> |-|-|-|-|
> | FedMef | 0.834 | 0.788| 0.784|
> | FedRTS | 0.838 | 0.792| 0.788|
>
> Notably, FedRTS consistently outperforms FedMef under different degrees of covariate shift, demonstrating its robustness in heterogeneous data-splitting settings.

---

> ### Author Response · Authors · 2025-08-04
> **Thank you for your review**
>
> Thank you for your valuable feedback and thorough review of our manuscript. We have carefully revised the paper to address all the raised concerns, including **adding further explanations, a complexity analysis, and a theoretical convergence analysis** in the updated version.
>
> **If our revisions meet your expectations, we would greatly appreciate it if you could consider increasing your rating.** Should you have any further questions or concerns, please do not hesitate to contact us. We would be happy to provide clarifications before the rebuttal period ends !

---

> > ### Comment · Reviewer_1eQR · 2025-08-05
> >
> > Thank you for your rebuttal. Most of my concerns have been addressed. However, I find the novelty of the paper somewhat below expectations, as the main ideas—such as Thompson sampling and federated pruning—have already been explored in prior work. The primary contribution of this paper appears to be an extension of these existing ideas to a specific scenario. Therefore, I have decided to maintain my original score.

---

> ### Author Response · Authors · 2025-08-05
> **Thanks for further comments**
>
> We sincerely appreciate the reviewer’s constructive comments and are glad that most of their concerns have been addressed. Regarding the novelty of our work, we respectfully highlight the following key contributions:
>
> - __Novel Problem Formulation__: ___We are the first to formulate federated dynamic pruning as a combinatorial multi-armed bandit problem, providing a principled theoretical framework for this task (as acknowledged by Reviewers aLmC, EiZ9, and UDMM).___ This represents a significant departure from existing heuristic approaches, enabling FedRTS to systematically overcome three fundamental limitations in current methods: greedy adjustments, unstable topologies, and communication inefficiency.
> - __Theoretical Advancements__: Our work introduces the first regret bound and non-convex convergence analysis for federated dynamic pruning, where prior work offered only empirical validation. Therefore, _the application of Thompson sampling in this context is NOT merely an extension but a systematic solution with theoretical guarantees._
>
> We hope these clarifications highlight how our work provides both foundational theoretical insights and practical algorithmic advances to this emerging research area. We respect the reviewer's perspective regarding the score and we greatly appreciate their time and valuable insights, which have significantly strengthened our paper.

---

### Official Review · Reviewer_UDMM · 2025-06-29

**Clarity:** 3
**Significance:** 3
**Originality:** 3
**Rating:** 4
**Confidence:** 3

**Summary:**

The goal of the paper is to propose a federated solution for robust model pruning. The key design of this manuscript is the proposed FedRTS, which can produce robust sparse models via combinatorial Thompson sampling. The author claims that the proposed FedRTS improves robustness and performance using a Thompson Sampling-based Adjustment (TSAdj) mechanism and addresses three critical issues of existing approaches—greedy adjustments, unstable topologies, and communication inefficiency. Authors evaluate proposed methods using Lightweight CV models (ResNet18, ShuffleNetV2) and one small NLP model (GPT-2 32M) on TinyStories.

**Questions:**

Typo: Rubust -> Robust (L45)

Authors may compare the relevant resource use of different methods. Ideally, just performance gain is not enough for real-world distributed ML. To show the practicability of the proposed method in a real-world setting, authors may provide more details about resource usage (communication/computation cost). In the proposed method, besides the standard FL stage, Thompson Sampling-based Adjustment may introduce an overhead of computation and reduce the practicality in the real-world deployment. The author may update Table 1 to include a comparison of computation costs.

**Ethical Concerns:**

["NO or VERY MINOR ethics concerns only"]

**Limitations:**

yes

**Quality:**

3

**Strengths And Weaknesses:**

Strengths:

1. This is an interesting work with notable results in terms of performance in different settings. I also appreciated the efforts to evaluate the methods, notably the experiment settings (Sec. 4.2). This work addressed an important problem in the field, and so will be of interest to the readership of the conference.

2. The paper is well-written and easy to follow.

3. A novel Thompson Sampling-based Adjustment (TSAdj) mechanism is proposed, and the methodology is evaluated via multiple publicly available datasets.

Weaknesses:

1. Assumptions in theoretical analysis may not hold in practice. The theoretical analysis (Section 3.3, Theorem 3.4) relies on several simplifying assumptions. For example, Assumption 3.2. (Independent importance score) The importance scores used for pruning are treated as mutually independent. However, in real neural networks, weight dependencies exist, and such independence or approximations may not hold, especially in small models or sensitive layers (e.g., attention heads in NLP).

2. Design choices for hyperparameters are heuristic. Several components, such as the trade-off parameter γ in fusion, the scaling factor λ, and the number of core links κ, are chosen empirically or based on prior works. Though some tuning is shown in Appendix E, there is no systematic analysis of how sensitive FedRTS is to these choices. This can affect reproducibility and robustness in other settings, especially if applied to unseen tasks or models.

---

> ### Author Rebuttal · Authors · 2025-07-31
>
> We sincerely thank Reviewer UDMM for the detailed review and valuable feedback on our manuscript. We are encouraged that they found our work interesting and appreciated that the paper is well-written and easy to follow. Below, we provide detailed responses to all questions.
>
>
> > **W1** Assumptions in theoretical analysis may not hold in practice. **The theoretical analysis (Section 3.3, Theorem 3.4) relies on several simplifying assumptions.** For example, Assumption 3.2. (Independent importance score) The importance scores used for pruning are treated as mutually independent.
>
> We acknowledge that the assumption of independent importance scores is a simplification. However, this assumption stems from the nature of __magnitude-based pruning__, rather than our regret analysis itself. Prior pruning methods [1,2] commonly rank individual weight magnitudes to assign importance, and treating these scores as independent has become a standard practice.
> Moreover, while weight magnitudes are not strictly independent, both theoretical results and empirical evidence suggest that __trained weights exhibit asymptotic independence__ [3,4], lending support to this assumption.
>
> Importantly, our Theorem 3.4 is __agnostic to the specific pruning technique__; it holds for __any pruning method__ satisfying Assumptions 3.2 and 3.3, including future methods that generate independent importance scores more efficiently.
>
> [1] Song Han, Huizi Mao, and William J Dally. Deep compression: Compressing deep neural net works with pruning, trained quantization and huffman coding. arXiv preprint arXiv:1510.00149, 355 2015
>
> [2] Xin Qian and Diego Klabjan. A probabilistic approach to neural network pruning. In International Conference on Machine Learning, pages 8640–8649. PMLR, 2021.
>
> [3] Justin Sirignano and Konstantinos Spiliopoulos. Mean field analysis of neural networks: A central limit theorem. Stochastic Processes and their Applications, 130(3):1820–1852, 2020.
>
> [4]  Justin Sirignano and Konstantinos Spiliopoulos. Mean field analysis of neural networks: A law of large numbers. SIAM Journal on Applied Mathematics, 80(2):725–752, 2020.
>
> > **W2** Design choices for hyperparameters are heuristic. Several components, such as the trade-off parameter γ in fusion, the scaling factor λ, and the number of core links κ, are chosen empirically or based on prior works. Though some tuning is shown in Appendix E, there is no systematic analysis of how sensitive FedRTS is to these choices. This can affect reproducibility and robustness in other settings, especially if applied to unseen tasks or models.
>
> We appreciate the reviewer’s insightful feedback regarding the empirical selection of hyperparameters and we acknowledge that the hyperparameter values tested in our ablation studies (Appendix E) were selected based on empirical priors from related works and initial exploratory experiments rather than through a comprehensive systematic approach (e.g., grid search over continuous parameter spaces). Despite this method, we ensured our parameter selection process maintained full transparency and reproducibility. Furthermore, we conducted systematic sensitivity analyses for each parameter as detailed below:
> * $\gamma$ controls the balance between individual and aggregated information: a large $\gamma$ places greater emphasis on aggregated information, while a small  $\gamma$ concentrates more on individual outcomes.
> * $\lambda$ scales the evidence added to the beta distributions' parameters ($\alpha$, $\beta$), which directly affect link selection: a large $\lambda$ results in large increments to $\alpha$ and $\beta$ for selected links, accelerating the convergence of their distributions and promoting exploitation of current high-performing links. Conversely, a small $\lambda$ applies small increments, maintaining higher uncertainty for less frequently selected links and encouraging exploration.
> * $\kappa$ determines the layer-wise pruning rate: a large $\kappa$ leads to slower topology evolution, while a small $\kappa$ risks substantial information loss.
> * $\Delta T$ impacts the frequency of topology adjustments: a large $\Delta T$ may delay the convergence to stable topology, while a small $\Delta T$ leads to frequent adjustments incurring higher resource costs.
>
> While our current analysis employs discrete sampling on hyperparameters' impacts, the conducted sensitivity analysis and empirical validation confirm FedRTS’s robustness and reproducibility. To directly address the reviewer's concern, we will implement systematic methods including grid search to comprehensively analyze FedRTS's sensitivity to these hyperparameters. This will be accompanied by:
> * A sensitivity heatmap quantifying performance variance across hyperparameters
> * Full publication of all hyperparameter configurations used in our experiments for reproducing
>
>
> > **Q1** Typo Rubust -> Robust (L45).
>
> Thanks for pointing this out, we will revise it in the final version.
>
>
> > **Q2** In the proposed method, besides the standard FL stage, Thompson Sampling-based Adjustment may introduce an overhead of computation and reduce the practicality in the real-world deployment. The author may update Table 1 to include a comparison of computation costs.
>
> We sincerely appreciate the reviewer's valuable feedback highlighting the necessity of resource cost evaluation for real-world applicability, especially concerning the potential overhead of TSAdj. We fully agree that performance gains must be evaluated alongside resource usage.
>
> In TSAdj's design, all computations occur exclusively on the server, and clients incur zero additional FLOPs compared to baselines (FedDST/FedMef/FedTiny/PruneFL). The server-side FLOPs per outer loop include: global aggregation ($\mathcal{O}( \langle C_t \rangle \langle W \rangle)$), outcome updating ($\mathcal{O}(\langle C_t \rangle K)$), distribution updates ($\mathcal{O}(\langle W \rangle)$), beta sampling ($\mathcal{O}(\langle W \rangle)$), and mask adjustment ($\mathcal{O}(\langle C_t \rangle (\langle W \rangle - K) + \langle W \rangle)$).
>
> Therefore, we can simplify the computational complexity to $\mathcal{O}(\langle C_t \rangle \langle W \rangle)$ as $K = d \langle W \rangle$. Compared to the baselines' complexity $\mathcal{O}(\langle C_t \rangle \langle W \rangle)$ (global aggregation and mask adjustment), our proposed TSAdj achieves the same computational complexity, demonstrating that TSAdj introduces no significant additional computational burden on the server.
>
> **Most importantly, TSAdj introduces no additional computation on devices compared to the baselines**. Its overhead is entirely server-side. In cross-device FL, the server is typically assumed to have substantial computational resources; therefore, TSAdj does not impose a significant extra burden compared to the baselines. Below is the revised Table 1 with on-device FLOPs analysis:
>
>
> |Target Density|Method|PPL| FLOPs|Comm. Cost|
> |-|-|-|-|-|
> |1|FedAVG|20.56|5.76E+08|260.41 MB|
> |50%|FedDST|20.10|1.24E+08|138.30 MB|
> |50%|FedSGC|26.13|1.28E+08|207.45 MB|
> |50%|FedMef|20.61|1.22E+08|165.96 MB|
> |50%|**FedRTS**|18.54|1.24E+08|138.84 MB|
> |30%|FedDST|24.46|3.89E+07|86.26 MB|
> |30%|FedSGC|32.39|3.86E+07|129.39 MB|
> |30%|FedMef|21.00|4.14E+07|103.51 MB|
> |30%|**FedRTS**|18.56|3.89E+07|86.44 MB|
> |20%|FedDST|26.49|2.00E+07|60.22 MB|
> |20%|FedSGC|43.00|2.17E+07|90.33 MB|
> |20%|FedMef|21.53|1.99E+07|72.26 MB|
> |20%|**FedRTS**|19.93|2.00E+07|60.34 MB|

---

### Official Review · Reviewer_EiZ9 · 2025-07-03

**Clarity:** 3
**Significance:** 1
**Originality:** 2
**Rating:** 5
**Confidence:** 2

**Summary:**

This paper reframes federated dynamic pruning as a combinatorial multi-armed bandit (CMAB) problem. The key idea is to replace the deterministic, greedy topology updates used in prior work with a Thompson-Sampling-based Adjustment (TSAdf). This makes pruning decisions probabilistic, history-aware and communication-efficient (only largest gradients). The authors derive an upper bound on the regret. RedRTS either improves accuracy up to 5.1pp or reduces communication by 33% on vision and NLP benchmarks.

**Questions:**

- Pruning is mainly for large models rather than ResNet18. Given that federated learning is evaluated even on LLMs, please evaluate on large models that can show the impact of pruning explicitly.

- Magnitude-based pruning is the most traditional method. Other methods, such as Fisher-information, can outperform but were simply deemed due to their computational burden. Can you empirically quantify the pros and cons of these methods?

- Assumption 3.2 (independence between weight magnitude) is not true, which needs to be justified clearly.

**Ethical Concerns:**

["NO or VERY MINOR ethics concerns only"]

**Final Justification:**

The rebuttal has addressed my concerns successfully.

**Limitations:**

Please state the limitations of your work in Conclusion to facilitate further work.

**Quality:**

3

**Strengths And Weaknesses:**

**Strengths**
- Utilizing Thompson sampling is reasonable and new in the FL literature
- Theoretical regret analysis is provided with proof.
- Evaluation on both computer vision and NLP tasks
- Superior accuracy with less communication cost

**Weaknesses**
- Lack of evaluation on modern large models that shows the impact of pruning more clearly.
- Magnitude-based pruning is applied but other metric, such as Fisher-information, would be good to be compared empirically.
- Assumption 3.2 (independence between weight magnitude) incurs discrepancy between practice and theory.

---

> ### Author Rebuttal · Authors · 2025-07-31
>
> We sincerely appreciate the reviewer's detailed feedback, which has guided us to strengthen our work. We’ve addressed your concerns: evaluating pruning on large models (via OPT-125M experiments), comparing magnitude-based pruning with Fisher-information, and clarifying Assumption 3.2’s independence. Below, we detail these efforts to enhance our paper’s rigor.
>
> > **W1 & Q1** Lack of evaluation on modern large models that shows the impact of pruning more clearly.
>
> We have conducted additional experiments on the OPT-125M model using the OIG/Chip2 dataset and evaluated it on the last-word prediction task in Lambada. To balance cost and feasibility, we used 100 clients, selecting 8 clients per round, with a target density of 0.5.
>
> |OPT-125M | 	Acc | Comm|
> |--|-|-|
> |FedAVG|	0.4311|	 1.05 GB|
> |FedMef|	0.4307 |	0.59 GB|
> |FedRTS|	0.4315 |	0.55 GB|
>
> These results demonstrate that FedRTS maintains higher accuracy with lower communication cost on LLM. We note that cross-device FL is seldom used to fully train LLMs due to the extremely high cost on local devices.
>
>
> > **W2 & Q2**  Magnitude-based pruning is applied but other metrics, such as Fisher-information, would be good to be compared empirically.
>
> We appreciate the reviewer's suggestion to evaluate Fisher-information-based pruning versus magnitude-based pruning (FedRTS). While the Fisher-based approach empirically achieves 0.1% higher accuracy by approximating the Hessian inverse for better parameter importance estimation [1], it introduces prohibitive overheads in the FL setting.
>
> Specifically, Fisher-based pruning requires dense gradient computations (e.g., multiplications) because it employs the Sherman-Morrison formula to approximate direct inverse Hessian operations, which would otherwise exhibit cubic complexity in the number of parameters. However, this approach still maintains quadratic $\mathcal{O}(\langle W \rangle^2)$ complexity per batch - substantially higher than FedRTS's linear $\mathcal{O}(\langle W \rangle)$ operations. The detailed formula of the empirical Fisher is:
>
> $$\hat{F_{n+1}^{-1}} = \hat{F_n^{-1}} - \frac{\hat{F_n^{-1}}\nabla l_{n+1}\nabla l_{n+1}^T\hat{F_n^{-1}}}{N + \nabla l_{n+1}^T\hat{F_n^{-1}}\nabla l_{n+1}}$$
>
> Furthermore, a good approximation of the empirical Fisher needs a large number of samples for recurrent computation which is mentioned in Fisher approximation. Hence, its computational cost approaches that of dense FedAVG, which directly contradicts the core objective of federated pruning - reducing resource consumption especially on devices. Therefore, magnitude-based pruning (FedRTS) achieves an efficiency-accuracy balance in cross-device FL's setting.
>
> |Method| Acc | FLOPs |
> |--|-|-|
> |FedRTS (Fisher-information-based pruning) |0.7112| 1.31E+13 |
> |FedRTS (Magnitude-based pruning) |0.710225| 3.82E+12  |
>
> [1] Singh S P, Alistarh D. Woodfisher: Efficient second-order approximation for neural network compression[J]. Advances in Neural Information Processing Systems, 2020, 33: 18098-18109.
>
> > **W3 & Q3**  Assumption 3.2 (independence between weight magnitude) incurs discrepancy between practice and theory.
>
> We acknowledge that Assumption 3.2 (independent importance scores) is a simplification. However, this stems from the nature of magnitude-based pruning, a standard practice in prior federated pruning methods [2, 3] .
>
> While strict independence is not absolute, theoretical and empirical evidence shows that trained weights in large networks exhibit asymptotic independence [4, 5], supporting this approximation.
>
> Notably, our Theorem 3.4 is agnostic to specific pruning techniques; it holds for any method satisfying Assumptions 3.2 and 3.3, including future advancements in generating independent scores.
>
> [2] Yuang Jiang, Shiqiang Wang, Victor Valls, Bong Jun Ko, Wei-Han Lee, Kin K Leung, and Leandros Tassiulas. Model pruning enables efficient federated learning on edge devices. IEEE Transactions on Neural Networks and Learning Systems, 34(12):10374–10386, 2022.
>
> [3] Hong Huang, Weiming Zhuang, Chen Chen, and Lingjuan Lyu. Fedmef: Towards memory-efficient federated dynamic pruning. In Proceedings of the IEEE/CVF Conference on Computer Vision and Pattern Recognition, pages 27548–27557, 2024.
>
> [4] Justin Sirignano and Konstantinos Spiliopoulos. Mean field analysis of neural networks: A central limit theorem. Stochastic Processes and their Applications, 130(3):1820–1852, 2020.
>
> [5] Justin Sirignano and Konstantinos Spiliopoulos. Mean field analysis of neural networks: A law of large numbers. SIAM Journal on Applied Mathematics, 80(2):725–752, 2020.
>
>
> > **Limitations** Please state the limitations of your work in Conclusion to facilitate further work.
>
> We thank the reviewer for this valuable suggestion. We will state the limitations of our proposed FedRTS in Conclusion of the revised version to explicitly discuss the boundaries of our work and facilitate further work.

---

> > ### Comment · Reviewer_EiZ9 · 2025-08-09
> > **Thanks for the rebuttal**
> >
> > Thanks for the efforts. My concerns have been addressed and I will keep the positive score.

---

### Official Review · Reviewer_aLmC · 2025-07-06

**Clarity:** 3
**Significance:** 4
**Originality:** 1
**Rating:** 5
**Confidence:** 4

**Summary:**

This paper proposes FedRTS, a novel federated pruning framework that improves the robustness, efficiency, and adaptability of sparse model training in federated learning (FL) environments. At the core of FedRTS is the Thompson Sampling-based Adjustment (TSAdj) mechanism, which formulates topology adjustment as a combinatorial multi-armed bandit problem. Rather than relying on deterministic and short-sighted decisions, TSAdj makes probabilistic, history-aware selections of model connections based on both local (client) and global (server-aggregated) outcomes.

The framework introduces a fusion mechanism that blends individual and aggregated observations to mitigate instability and variance in sparse topology updates. Additionally, it reduces communication overhead by requiring clients to upload only top-gradient indices instead of full updates.

FedRTS is evaluated on both computer vision (e.g., CIFAR-10 with ResNet18) and natural language processing (TinyStories with GPT-2) tasks under various sparsity levels, data heterogeneity, and client availability conditions. Across all settings, it consistently outperforms existing sparse FL baselines in terms of accuracy, communication cost, and robustness. An ablation study further confirms that the gains stem from the probabilistic adjustment strategy, use of farsighted information, and the stability offered by the fusion mechanism.

**Questions:**

Computational Overhead of TSAdj
While FedRTS improves communication efficiency, TSAdj maintains and samples from Beta distributions for each model link, which could be expensive for large-scale models (e.g., GPT-2).
Question: Can the authors provide a computational complexity analysis or empirical runtime comparison (e.g., per round) against baseline methods?
Evaluation impact: Addressing this would clarify scalability concerns and reinforce the practicality of FedRTS in large models.

Validation of Inactive Link Outcome Estimation (Eq. 5)
The estimation of outcomes for inactive links via gradient magnitudes is a key part of reducing communication cost, but its reliability is not rigorously evaluated.
Question: How sensitive is TSAdj's performance to this heuristic? Can the authors quantify its accuracy or compare it with alternative estimators (e.g., partial gradient transmission)?
Evaluation impact: A justification or ablation could strengthen confidence in the outcome modeling and the robustness of FedRTS.

Hyperparameter Sensitivity and Tuning
The paper briefly mentions that hyperparameters (γ, λ, κ, ∆T) affect performance, but defers analysis to the appendix.
Suggestion: Please summarize key findings on sensitivity (e.g., which parameters are most impactful, how robust is the method to mis-tuning) in the main paper.
Evaluation impact: Improved clarity here would support broader applicability and ease of adoption.

Real-world FL Setting or Simulation Realism
The current evaluation is thorough but confined to synthetic federated setups.
Suggestion: Could the authors comment on (or ideally include) experiments in more realistic FL settings (e.g., device-level heterogeneity, on-device constraints)?
Evaluation impact: Demonstrating FedRTS under deployment-like conditions would significantly strengthen the practical significance of the work.

Broader Applicability Beyond Pruning
FedRTS is positioned as a pruning method, but the TSAdj mechanism could, in principle, generalize to other structural decisions in FL.
Question: Do the authors envision applications of TSAdj beyond pruning (e.g., layer-wise selection, gradient sparsification)? If so, a brief discussion would be valuable.

**Ethical Concerns:**

["NO or VERY MINOR ethics concerns only"]

**Final Justification:**

The authors have added experiments to address my concerns. It is a strong paper now.

**Limitations:**

The authors have not explicitly discussed the limitations or potential negative societal impacts of their work. While the technical contributions are strong, the paper would benefit from a short section or paragraph acknowledging:

Scalability limitations: TSAdj may introduce overhead when scaling to very large models with millions of parameters. A discussion of this trade-off is important for deployment feasibility.
Fairness and bias: The method dynamically adjusts model structure per global feedback. In highly heterogeneous settings, this could unintentionally bias the model toward data-rich or majority-class clients.
Security/privacy considerations: While the method reduces communication cost, sharing top gradient indices might still leak sensitive information. A brief mention of this risk and possible mitigations (e.g., differential privacy) would be constructive.
Suggestion: Add a limitations section addressing scalability, bias in client selection/topology adaptation, and communication-related privacy concerns. This would strengthen the paper’s transparency and societal responsibility.

**Paper Formatting Concerns:**

No issues found.

**Quality:**

4

**Strengths And Weaknesses:**

Strengths

Originality: Novel framing of federated pruning as a combinatorial bandit problem; innovative use of Thompson Sampling for sparse topology adjustment.
Quality: Strong theoretical foundation (regret bound), well-designed TSAdj algorithm, and comprehensive experiments across modalities (CV/NLP) and FL challenges (non-IID, partial participation).
Significance: Addresses a core bottleneck in sparse FL—instability and communication inefficiency—with clear improvements over strong baselines (FedAVG, FedDST, FedMef).
Clarity: Method and motivations are well-explained; algorithm steps, equations, and experiment setup are easy to follow. Ablation study isolates contributions effectively.

Weaknesses

Computational cost of maintaining and sampling Beta distributions for large models is not analyzed.
Outcome estimation for inactive links relies on heuristic (gradient magnitude) without deeper justification or validation.
Lack of real-world deployment or large-scale FL setting; all evaluations are in simulated environments.
Hyperparameter sensitivity (e.g., γ, λ) is deferred to the appendix, limiting insight into robustness in the main text.

---

> ### Author Rebuttal · Authors · 2025-07-31
>
> We sincerely appreciate the reviewer aLmC's __detailed and insightful feedback__ and we are encouraged by the recognition of our paper’s strong theoretical foundation, well-designed algorithm, comprehensive experiments, and clear explanations. We have carefully addressed the concerns in detail in the responses below to enhance clarity and refinement.
>
> > **Q1 & W1** Computational cost of maintaining and sampling Beta distributions for large models is not analyzed.
>
> In TSAdj's design, the server performs the following operations at each round:
> weights aggregation iterates exactly $\langle C_t \rangle$ times, requiring $\mathcal{O}(\langle C_t \rangle \langle W \rangle)$ for weight averaging, $\mathcal{O}(\langle C_t \rangle \langle W \rangle)$ for outcome computation (expected time with Quick Sort), $\mathcal{O}(\langle W \rangle)$ for updating beta distributions and sampling probabilities, $\mathcal{O}(\langle W \rangle) $ for arms selection.
>
> The overall complexity simplifies to  $\mathcal{O}(\langle C_t \rangle \langle W \rangle)$.
>
> Compared to baselines' complexity (e.g., FedMef/FedTiny/PruneFL's $\mathcal{O}(\langle C_t \rangle \langle W \rangle)$), our proposed TSAdj achieves the same computational complexity order, demonstrating that TSAdj introduces no significant additional computational burden on the server.
>
> **Most importantly, TSAdj introduces no additional computation on devices compared to the baseline**. Its overhead is entirely server-side. In cross-device FL, the server is typically assumed to have substantial computational resources; therefore, TSAdj does not impose a significant extra burden compared to the baselines.
>
> > **Q2.1 & W2**  Outcome estimation for inactive links relies on heuristic (gradient magnitude) without deeper justification or validation.
>
> Our use of gradient magnitude for estimating outcomes of inactive links (Eq. 5) is motivated by both empirical precedent and theoretical intuition:
> 1.Empirical: Gradient magnitude is a key metric in dynamic sparse training. Methods like FedTiny, FedMef, and RigL focus on parameters with large gradients for reactivation, ensuring fair comparison.
> 2. Theoretical: Large gradient magnitudes indicate a parameter's importance for model optimization. Keeping parameters with high gradients can notably reduce loss, aligning with pruning theory's "saliency" principle.
>
> However, we admit this heuristic and may not capture complex parameter interactions, we will explore theoretically grounded alternatives in future work.
>
> [1] Evci, Utku, et al. "Rigging the lottery: Making all tickets winners." International conference on machine learning. PMLR, 2020.
>
>
> > **Q2.2** Can the authors quantify its accuracy or compare it with alternative estimators (e.g., partial gradient transmission)
>
> We compared FedRTS with FedTiny using both partial and full gradient transmission on CIFAR-10. The results show that FedRTS achieves higher accuracy than FedTiny with full gradient transmission, while incurring lower communication cost than FedTiny with partial gradient transmission. This is because FedRTS only needs to transmit a binary (0/1) mask matrix instead of the gradient values.
>
> || Acc | Comm. Cost |
> |--|-|-|
> | FedTiny (partial gradient) |0.554| 25.32 MB   |
> |  FedTiny (full gradient)   |0.573| 55.19 MB   |
> | FedRTS|0.580| 20.85 MB   |
>
> > **Q3 & W4**  Please summarize key findings on sensitivity (e.g., which parameters are most impactful, how robust is the method to mis-tuning) in the main paper.
>
> We will add a new subsection in Section 4.3, summarizing key findings on hyperparameter sensitivity.
> * $\gamma$ controls the balance between individual and aggregated information: a large $\gamma$ places greater emphasis on aggregated information, while a small  $\gamma$ concentrates more on individual outcomes.
> * $\lambda$ scales the evidence added to the beta distributions' parameters ($\alpha$, $\beta$), which directly affect link selection: a large $\lambda$ results in large increments to $\alpha$ and $\beta$ for selected links, accelerating the convergence of their distributions and promoting exploitation of current high-performing links. Conversely, a small $\lambda$ applies small increments, maintaining higher uncertainty for less frequently selected links and encouraging exploration.
> * $\kappa$ determines the layer-wise pruning rate: a large $\kappa$ leads to slower topology evolution, while a small $\kappa$ risks substantial information loss.
> * $\Delta T$ impacts the frequency of topology adjustments: a large $\Delta T$ may delay the convergence to stable topology, while a small $\Delta T$ leads to frequent adjustments incurring higher resource costs.
>
> The most impactful parameters are $\gamma$ and $\lambda$ as accuracy varies $\pm 1.2$% for $\gamma \in [0, 1]$ and accuracy peaks at $\lambda = 10$ ($+1.8$% v.s. $\lambda = 1$). However, FedRTS still outperforms other baselines even with suboptimal hyperparameters based on the ablation study, demonstrating the robustness to mis-tuning.
>
> > **Q4 & W3** The current evaluation is thorough but confined to synthetic federated setups. Could the authors comment on experiments in more realistic FL settings?
>
> We plan to include real-world FL deployments in future work. Here, we briefly discuss the expected effects and dependencies of FedRTS’s unstructured sparsification in practical settings: __1. Communication Efficiency:__ FedRTS’s sparse updates can be readily compressed using standard sparse storage schemes (e.g., CSR/CSC, BitMap), achieving similar communication cost savings as reported in this paper, without hardware dependencies. __2. Memory/Storage Efficiency:__ The sparse models generated by FedRTS can leverage existing sparse ML engines (e.g., PyTorch’s torch.sparse_coo_tensor, TinyML) to reduce memory usage. For example, MCUNet [2] reports a 3.8× memory reduction for 90%-sparse models using TinyML. 3. __Computation Efficiency:__ Latency gains from unstructured sparsity are hardware-dependent. For instance, Jiang et al. [3] demonstrate 1.5–2.5× training speedups on Raspberry Pi 4 using sparse kernels.
>
> [2] Lin, Ji, et al. "Mcunet: Tiny deep learning on iot devices." Advances in neural information processing systems 33 (2020): 11711-11722.
>
> [3] Jiang, Yuang, et al. "Model pruning enables efficient federated learning on edge devices." IEEE Transactions on Neural Networks and Learning Systems 34.12 (2022): 10374-10386.
>
> > **Q5** Do the authors envision applications of TSAdj beyond pruning (e.g., layer-wise selection, gradient sparsification)?
>
> Yes, the TSAdj mechanism can generalize to __any decision problem requiring adaptive, long-term optimization under uncertainty__. For instance, TSAdj could be applied to layer-wise selection or gradient sparsification by dynamically activating/deactivating layers or gradients based on the maintained Beta distributions. A consideration, as the reviewer noted earlier, is that maintaining Beta distributions introduces some overhead. Therefore, TSAdj is best suited for distributed settings, such as FedRTS, where the cost can be centralized to a powerful host or parallelized to clients. We view these extensions as exciting future directions and welcome further exploration.
>
> > **Limitation**  The authors have not explicitly discussed the limitations or potential negative societal impacts of their work.
>
> We will include the following limitations in the paper:
>
> 1. __Scalability:__ TSAdj maintains Beta distributions for parameter-level decisions, which may introduce additional overhead when scaling to extremely large models with millions of parameters on the server. Although servers typically have substantial resources and the computational complexity remains comparable to baseline methods, further optimizations (e.g., hierarchical grouping or parameter sharing) are needed for deployment on very large models.
>
> 2. __Fairness and Bias:__ Because TSAdj dynamically adjusts the model structure based on global feedback, in highly heterogeneous settings, this mechanism could unintentionally bias the model toward data-rich clients or majority-class distributions. Future work will explore fairness-aware adjustments to mitigate such risks.
>
> 3. __Security and Privacy:__ While FedRTS reduces communication costs by transmitting only sparse updates (e.g., top gradient indices), this information could still leak sensitive details about client data. Integrating techniques such as differential privacy or secure aggregation into FedRTS could strengthen privacy guarantees.

---

### Author Response · Authors · 2025-08-04
**General Response to All Reviewers**

We sincerely appreciate all the reviewers for their time and constructive feedback on our manuscript. We are particularly encouraged that: Reviewer aLmC recognized the strong theoretical foundation of our work;  Reviewer 1eQR think our experiments are very comprehensive; Reviewer EiZ9 found our idea both reasonable and new; Reviewer UDMM considered our work interesting and well-written.

We have carefully addressed every question and concern raised by all four reviewers. This involved conducting additional experiments as well as performing further theoretical convergence analysis. All content/experiments/proof in responses have been included in the updated manuscript. We do hope that our response will dissipate the doubts concerning certain parts of the method.

---

### Decision · Program_Chairs · 2025-09-17

**Decision:**

Accept (poster)

**Comment:**

This paper aims to enhance existing federated pruning methods through probabilistic history-aware decisions based on a Thompson-sampling adjustment. Casting federated pruning as a combinatorial bandit problem is novel. Clear improvements over baselines are demonstrated, and a theoretical regret analysis with proof is provided.